# Controlling hot electron flux and catalytic selectivity with nanoscale metal-oxide interfaces

Si Woo Lee [1,2,7], Jong Min Kim[3,4,7], Woonghyeon Park[5,7], Hyosun Lee[1,6], Gyu Rac Lee[3], Yousung Jung [5✉], Yeon Sik Jung [3✉] & Jeong Young Park [1,2✉]

Interaction between metal and oxides is an important molecular-level factor that influences the selectivity of a desirable reaction. Therefore, designing a heterogeneous catalyst where metal-oxide interfaces are well-formed is important for understanding selectivity and surface electronic excitation at the interface. Here, we utilized a nanoscale catalytic Schottky diode from Pt nanowire arrays on $TiO_2$ that forms a nanoscale Pt-$TiO_2$ interface to determine the influence of the metal-oxide interface on catalytic selectivity, thereby affecting hot electron excitation; this demonstrated the real-time detection of hot electron flow generated under an exothermic methanol oxidation reaction. The selectivity to methyl formate and hot electron generation was obtained on nanoscale Pt nanowires/$TiO_2$, which exhibited ~2 times higher partial oxidation selectivity and ~3 times higher chemicurrent yield compared to a diode based on Pt film. By utilizing various Pt/$TiO_2$ nanostructures, we found that the ratio of interface to metal sites significantly affects the selectivity, thereby enhancing chemicurrent yield in methanol oxidation. Density function theory (DFT) calculations show that formation of the Pt-$TiO_2$ interface showed that selectivity to methyl formate formation was much larger in Pt nanowire arrays than in Pt films because of the different reaction mechanism.

[1] Center for Nanomaterials and Chemical Reactions, Institute for Basic Science (IBS), Daejeon 34141, Republic of Korea. [2] Department of Chemistry, Korea Advanced Institute of Science and Technology (KAIST), Daejeon 34141, Republic of Korea. [3] Department of Materials Science and Engineering, Korea Advanced Institute of Science and Technology (KAIST), Daejeon 34141, Republic of Korea. [4] Materials Architecturing Research Center, Korea Institute of Science and Technology (KIST), Seoul 02792, Republic of Korea. [5] Department of Chemical and Biomolecular Engineering, Korea Advanced Institute of Science and Technology (KAIST), Daejeon 34141, Republic of Korea. [6] Present address: Korea Institute of Industrial Technology (KITECH), Intelligent Sustainable Material R&D Group, Cheonan 31056, Republic of Korea. [7] These authors contributed equally: Si Woo Lee, Jong Min Kim, Woonghyeon Park. ✉email: ysjn@kaist.ac.kr; ysjung@kaist.ac.kr; jeongypark@kaist.ac.kr

Understanding the mechanisms of energy dissipation and electronic excitation at solid–gas interfaces on heterogeneous catalyst is important for various energy conversion applications to improve energy efficiency[1–4]. In particular, through exothermic catalytic reactions on a metal catalyst surface, non-adiabatic energy dissipation leads to the flow of energetic electrons with an energy of 1–3 eV (i.e., reaction-induced hot electrons) from a chemical energy conversion process; this flow of electrons operates on a femtosecond time scale before atomic vibrations dissipate the energy by adiabatic process[5–9]. However, because of the extremely short lifetime by thermalization (i.e., within a few femtoseconds via electron–electron or electron–phonon scattering) and mean free path of hot electrons in metal catalysts, directly detecting catalytically excited hot electrons before relaxation has been challenging[3,8,10]. Recently, metal–semiconductor Schottky nanodiodes, composed of a metal catalyst film deposited on a semiconductor substrate, have emerged as a powerful strategy for quantitative real-time detection of hot electron transfer excited on catalyst surface by exothermic chemical reactions[11–13]. When the potential barrier formed at the metal–semiconductor junctions (i.e., the Schottky barrier) is lower than the energy of the chemically excited electrons (1–3 eV) by catalytic reaction, the Schottky barrier allows excited hot electrons to irreversibly transport through the metal–semiconductor junction. Once hot electrons arrive at the semiconductor by transfer, these electrons cannot go back to the metal catalyst, leading to the irreversible and one-way hot electron transfer from the metal catalyst to the semiconductor. Therefore, metal–semiconductor catalytic nanodiodes allow the quick capture of energetic hot electrons before thermalization; the detected current is known as chemicurrent[5,9,14].

In heterogeneous catalysis, one of the key strategies for enhancing the reaction rate and selectivity is the use of catalytically active metal nanoparticles supported on an oxide support, forming a metal-oxide interface[15–17]. Strong metal-support interaction (SMSI) effect, in which the encapsulation of metal nanoparticles by reducible oxide-support overlayers can affect the catalytic behavior of metal nanoparticles (i.e., migration of oxide support onto the active metal surface) was discovered by Tauster et al.[18,19]. Matsubu et al.[20] also showed that adsorbate-functionalized encapsulation of metal nanoparticles by the reducible oxide support (e.g., $TiO_2$ and $Nb_2O_5$) can influence the selectivity of $CO_2$ hydrogenation in heterogeneous Rh catalysts (i.e., adsorbate-mediated SMSI effect) and these results were proved by observing changes in pre-treatment through in situ diffuse-reflectance infrared Fourier transform spectroscopy and in situ scanning transmission electron microscopy (TEM)[20]. Studies of low-temperature CO oxidation on gold nanoparticles on a $TiO_2$ support catalyst exhibits that the metal-oxide interface along its perimeter can dramatically change the catalytic activity[21,22]. Therefore, it can be seen that the formation of metal-oxide interface in heterogeneous catalyst plays an important role in altering the catalytic reaction.

In contrast to one-path reactions such as CO or hydrogen oxidation, the research on selectivity for achieving the desired product molecule in multi-path reactions is a challenging issue; this is the ultimate goal of a heterogeneous catalyst for green chemistry[4,20,23–25]. Among the various multi-path reactions, the gas-phase methanol oxidation reaction, which produces $CO_2$ and methyl formate by full oxidation and partial oxidation of methanol, is an important transformation process for the conversion of energy and chemical synthesis[26]. The partial oxidation of methanol to methyl formate is an especially efficient and environmentally benign way of producing valuable chemicals. Furthermore, it has been demonstrated that the oxidation state or chemical composition of the metal catalyst can affect the

selectivity to methyl formate under methanol oxidation[27–29]. Recently, it was found that the quantity of hot electrons generated in the methanol oxidation reaction is affected by the selectivity of methyl formate production using a Pt film/$TiO_2$ catalytic nanodiode[30]. In addition, previous studies have revealed that the partial oxidation of methanol increases when the metal-oxide interfacial sites (i.e., Pt/FeO and Au/$TiO_2$) are formed by using theoretical calculation and surface science techniques[31,32]. Therefore, the smart design of heterogeneous catalysts can improve the selectivity of metal catalysts on reducible oxide supports (e.g., Pt nanoparticles or nanowires on oxide support substrate) by engineering the metal-oxide interface[8].

In this study, to demonstrate the effect of metal-oxide interface on selectivity and reaction-induced hot electrons (i.e., chemicurrent), we fabricated a Schottky nanodiode with Pt nanowires on a $TiO_2$ support forming nanoscale Pt-$TiO_2$ interface that was exposed on the gas-phase reaction environment. Using the fabricated Pt nanowires/$TiO_2$ catalytic nanodiode, we report the in situ detection of charge flow as steady-state chemicurrent generated by gas-phase catalytic methanol oxidation on the Pt nanowire catalysts supported on $TiO_2$. To find the dynamics of reaction-induced hot electrons on the metal-oxide interface, the results of partial oxidation selectivity and chemicurrent on Pt nanowires/$TiO_2$ were compared with the results of the Pt film/$TiO_2$ Schottky nanodiode whose metal-oxide interface was not exposed to the environment of the gas-phase chemical reaction. We showed that formed nanoscale Pt-$TiO_2$ interface gives rise to the higher selectivity of methyl formate, thereby hot electrons were excited much more on the Pt-$TiO_2$ interface, as the hot electron generation was affected by selectivity under methanol oxidation. Comparing the simulation results by density functional theory (DFT) from the two models of Pt nanorod on $TiO_2$(110) and Pt(111), we could support the increase in selectivity at the Pt-$TiO_2$ interface due to the difference of activation barrier.

## Results

**Detection of hot electrons on Pt nanowires/$TiO_2$ nanodiodes.** The metal-oxide interface was fabricated on the catalytic Schottky nanodiodes by depositing the two-dimensional Pt nanowire arrays on the surface of $TiO_2$ using lithography, thus creating a well-defined nanoscale interface between the platinum and the titanium dioxide. A previous study on lithographically fabricated Pt nanowire arrays on oxide support found strong oxide support dependence for both TOF and the activation energy of the CO oxidation reaction[33,34]. Thus, in this study, the Pt nanowire arrays were chosen to form the metal-oxide interface, because these nanowires can be deposited on the $TiO_2$, while sustaining the electrical connection for the fabrication of Schottky nanodevices when exposing the well-defined Pt-$TiO_2$ interface to the reaction environment. A scheme of the fabricated catalytic nanodiode is presented in Fig. 1a, where the two-dimensional Pt nanowire arrays were deposited on the thin-film $TiO_2$ surface, forming a Pt nanowires/$TiO_2$ catalytic nanodiode. The figure in the inset shows that methanol oxidation occurs at the metal-oxide interface formed by the deposition of Pt nanowires on a $TiO_2$ surface. The Schottky nanodevice was well-constructed, the width of the Pt nanowires was 20 nm, and the pitch of the nanowire arrays was 50 nm, as shown in scanning electron microscopy (SEM) image.

The energy band diagram for real-time detection of hot electron flux as steady-state chemicurrent under methanol oxidation on Pt nanowires/$TiO_2$ catalytic nanodiode is illustrated in Fig. 1b. Here, a Au/Ti film layer for the ohmic contact electrodes of the nanodiodes on both sides is deposited by electron beam evaporation. As a probe reaction, methanol oxidation showing two reaction products

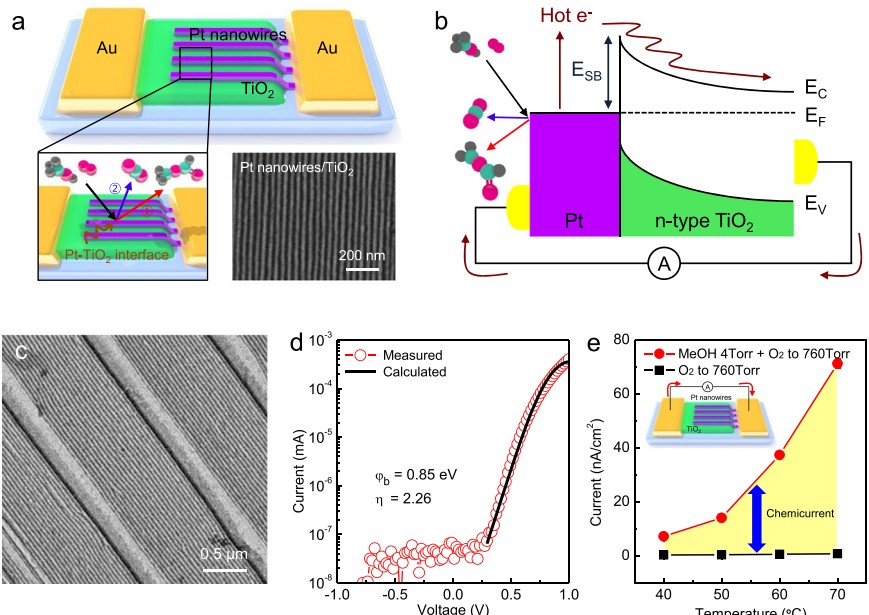

**Fig. 1 Detection of hot electrons generated on Pt nanowires/TiO₂ nanodiode. a** Schematic of hot electron generation on the Pt nanowires/TiO₂ catalytic nanodiode during exothermic methanol oxidation. Methanol oxidation occurs, and $CO_2$ and methyl formate are produced at the Pt-TiO₂ interface formed by the bonding of Pt nanowires and TiO₂. In this process, non-adiabatic energy dissipation occurs and hot electrons are generated on the nanodiode surface. SEM image of Pt nanowire arrays with a width of 20 nm and a pitch of 50 nm. **b** Energy band diagram for Schottky nanodiode of the Pt nanowires supported on TiO₂. Hot electrons excited from the surface chemical reaction can be detected as a steady-state chemicurrent by charge transfer if their excess chemical energy is large enough to overcome the Schottky barrier of the Pt-TiO₂ junction. **c** TEM image of the transferred Pt nanowire arrays on TEM grid with a width of 20 nm. **d** Current–voltage (*I*–*V*) curves for the Pt nanowires/TiO₂ catalytic nanodiode. The solid line is a fit of the obtained *I*–*V* curve to the thermionic emission theory. The obtained Schottky barrier height was 0.85 eV. **e** Current density associated with the methanol oxidation measured on the Pt nanowires/TiO₂ with increased reaction temperature. The differences in the magnitude of the currents measured with and without catalytic reaction were associated with reaction-induced hot electrons generated on the catalytic nanodiode (i.e., net chemicurrent).

(i.e., $CO_2$ and methyl formate production by full oxidation and partial oxidation of methanol reactant, respectively) was chosen, as it is an exothermic reaction with low temperature that is stable for maintaining the electrical properties of the catalytic nanodiode. When methanol oxidation, as an exothermic catalytic reaction, occurs in the catalyst, hot electrons were generated by non-adiabatic electronic excitation (i.e., chemical energy conversion) and only energetic electrons can be detected through the Schottky barrier formed at the metal–semiconductor junction as the electric current (i.e., chemical reaction-induced current flow).

To characterize the microstructure of Pt nanowires using TEM, the fabricated Pt nanowires were directly transferred onto a TEM grid in the same way that they were transferred to the nanodiodes for TEM sampling, as shown in Fig. 1c. Larger Pt nanowires with a width of 200 nm were observed between smaller Pt nanowires with a width of 20 nm. The formation of larger Pt nanowires was influenced by the topographical template required to control the long-range order of self-assembled block copolymer patterns[35,36]. Thus, we confirmed that the fabricated catalytic nanodiode was well designed, and that the junction formed between the Pt nanowire arrays and TiO₂ was stable.

To investigate the electrical properties and rectifying behavior of the fabricated Pt nanowires/TiO₂ catalytic nanodiode, current–voltage (*I*–*V*) curves were plotted. As shown in Fig. 1d, by fitting the *I*–*V* curve of the nanodiode to the thermionic emission equation, a Schottky barrier of 0.85 eV was obtained, which was similar to that of the device based on Pt film (see Supplementary Note 1 for details). Thus, the effect of Schottky barrier height on the hot electron detection can be excluded by showing a similar Schottky barrier to that of the Pt film (i.e., about 0.87 and 0.85 eV for Pt film/TiO₂ and Pt nanowires/TiO₂,

respectively). Recently, the detection of excited hot electrons as a steady-state chemicurrent under the catalytic oxidation of CO or hydrogen was demonstrated by using metal catalyst/TiO₂ Schottky nanodiodes and a clear correlation was found between catalytic activity and reaction-induced hot electron flux[10,37–41]. In this study, the chemically excited hot electrons generated by methanol oxidation were enough to irreversibly overcome the Schottky barrier at the Pt nanowire arrays/TiO₂ junction when the excess chemical energy was higher than the Schottky barrier height of the nanodiode (i.e., obtained sufficient energy) and could be detected as a current of hot electron flow.

The fabrication procedure of patterned Pt nanowire arrays on a TiO₂ support with self-assembled block copolymer is shown in Supplementary Fig. 1. Bulk Au nanowires, 200 nm, were printed on the Pt nanowires to prevent electrical shorts from defects in the Pt nanowires (Supplementary Fig. 2a). As the bulk Au nanowires were much larger than the Pt nanowire arrays, the Au nanowire arrays had no effect on the catalytic reaction. Moreover, X-ray photoelectron spectroscopy (XPS) was used to demonstrate the chemical composition of Pt nanowires on TiO₂, showing that the oxidation state was maintained in a metallic state (Supplementary Fig. 2b). In addition, X-ray diffraction (XRD) patterns of the Pt nanowires deposited on silicon wafer exhibited the polycrystalline nature of the transferred Pt nanowires (Supplementary Fig. 2c). To demonstrate the area density effect of the metal-oxide interface, nanodiodes were fabricated by depositing Pt nanowire arrays with widths of 15, 20, and 50 nm on a TiO₂ support; the morphology of these deposited Pt nanowire arrays and Pt film was confirmed by SEM (Supplementary Fig. 3a–d).

We monitored the electric current signals flowing across the Pt nanowires/TiO₂ Schottky contact at the open circuit by ammeter

during reaction conditions (methanol 1–4 Torr and $O_2$ to 760 Torr) and non-reactive conditions (pure $O_2$ 760 Torr) at elevated temperatures. The stable steady-state current signals on a Pt nanowires/$TiO_2$ Schottky nanodiode by catalytic methanol oxidation with elevated reaction temperatures are shown in Fig. 1e, where the measured electric current increased as the reaction activity increased with rising temperature. Under a pure oxygen environment (760 Torr of $O_2$), there was a weak thermoelectric current caused by the difference in electrical potential between the two electrodes (i.e., Seebeck effect)[41]. A definite deviation between the currents measured with and without catalytic reaction was observed and the difference in magnitude of the currents was clearly associated with hot electron generation from the catalytic methanol oxidation on the Pt-$TiO_2$ interface. Therefore, this result suggests that chemicurrent measurement by using the Pt nanowires/$TiO_2$ catalytic nanodiodes can indeed be used to monitor the surface chemical reaction at the metal-oxide interface in a quantitative and sensitive manner. Pt nanowires/$TiO_2$ catalytic nanodiodes exhibit good thermal stability, showing a stable steady-state chemicurrent due to electrical stability under oxygen-rich conditions (see Supplementary Note 2 and Supplementary Figs. 4–7 for details).

**Effect of metal-oxide interface on selectivity.** Recently, it was observed that selectivity of partial oxidation in the methanol oxidation reaction exhibited a significant increase when the Pt-$TiO_2$ interface was formed, which was well supported by theoretical calculations[42,43]. Thus, in this study, to find the metal-oxide interface effect on selectivity and reaction-induced hot electron, the results of Pt nanowire/$TiO_2$ were compared with a Pt film/$TiO_2$ without an exposed metal-oxide interface.

We calculated turnover frequency (TOF) for $CO_2$ and methyl formate production from the slope of the turnover number with reaction time carried out in a batch reactor system (Supplementary Fig. 8a). The selectivity to methyl formate formation was then calculated from the ratio of formation rate of methyl formate to $CO_2$; the selectivity to produce methyl formate decreases with increasing temperature, which is similar to the tendency in our previous results of Pt film/$TiO_2$ catalytic nanodiode (Supplementary Fig. 8b). The decline of selectivity with rising temperature was a result of the higher activation energy for full oxidation pathway than partial oxidation[26]. All TOF and selectivity were obtained in methanol conversion of less than 10%. As shown in Fig. 2a, the total TOF value (i.e., the sum of the TOFs of $CO_2$ and methyl formate production) was similar for Pt nanowires/$TiO_2$ and Pt film/$TiO_2$. Through this, it can be seen that the Pt-$TiO_2$ interface formed from Pt nanowires/$TiO_2$ does not affect the total reactivity. Unlike the TOF results, surprisingly, Pt nanowires/$TiO_2$ showed much higher selectivity toward methyl formate formation than Pt film/$TiO_2$; these results clearly indicate that the Pt-$TiO_2$ interface affects the selectivity of the chemical reaction, which is similar to previously reported results (Fig. 2b)[43]. Therefore, this indicates that when the Pt-$TiO_2$ interface was formed, $CO_2$ production was reduced and methyl formate production was increased, resulting in significant increase in selectivity, whereas total TOF was similar in both systems; thus, it can be said that Pt nanowires/$TiO_2$ was a more efficient catalyst for selective reaction producing methyl formate than Pt film/$TiO_2$. All the TOF values obtained from Pt film/$TiO_2$ and Pt nanowires/$TiO_2$, and the calculated selectivity are shown in Supplementary Tables 1 and 2, respectively. There have been previous studies to increase catalytic performance by increasing coverage degree of

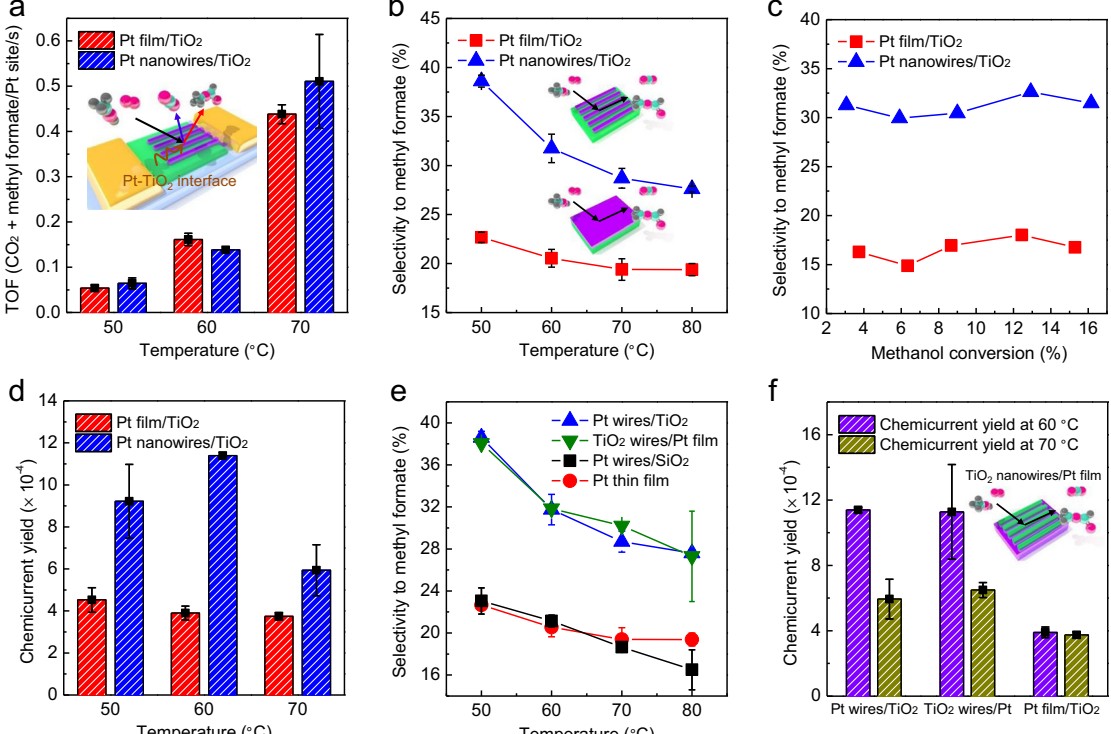

**Fig. 2 Selectivity and hot electron generation on nanodiodes.** Comparison of **a** total reactivity and **b** partial oxidation selectivity under methanol oxidation for two Schottky nanodiodes of Pt film and Pt nanowires supported on $TiO_2$ at different temperatures. **c** Plot of selectivity according to methanol conversion on the catalytic nanodiodes of Pt film/$TiO_2$ and Pt nanowires/$TiO_2$ measured at 343 K. **d** Chemicurrent yield associated with the efficiency for hot electron flow obtained from the Pt film/$TiO_2$ and Pt nanowires/$TiO_2$ catalytic nanodiodes during methanol oxidation reaction. **e** Selectivity to methyl formate and **f** chemicurrent yield under methanol oxidation on the Schottky nanodiodes with different oxide support or stack order of nanowires. All results were obtained under methanol 4 Torr and $O_2$ to 760 Torr.

metal nanocatalyst at interfacial sites when metal nanoparticles were supported by $TiO_2$[44–46]. In addition, earlier catalytic studies on two-dimensional Pt nanowire arrays deposited on reducible support (e.g., $CeO_2$ and $ZrO_2$) demonstrated increased catalytic activity on Pt nanowires with nanoscale metal-oxide interfaces than in Pt film or Pt(111) single crystal surface[33]. Therefore, in this study, we conclude that much-improved selectivity in our two-dimensional $TiO_2$ supported Pt nanowire arrays is associated with the nanoscale interfacial sites formed between Pt nanowires and $TiO_2$.

To more clearly show the selectivity differences and assess these differences on Pt nanowires/$TiO_2$ and Pt film/$TiO_2$, the selectivity according to methanol conversion at 343 K was plotted, as shown in Fig. 2c. We divided the regions for each section of conversion, calculated the TOF for each $CO_2$ and methyl formate production, then calculated selectivity and compared them in Pt nanowires/$TiO_2$ and Pt film/$TiO_2$. The selectivity of each catalyst according to methanol conversion was almost constant, which means that the reaction kinetics of methanol oxidation were constant within this comparable methanol conversion region. In addition, when comparing selectivity obtained from Pt nanowires/$TiO_2$ and Pt film/$TiO_2$, it was found that Pt nanowires/$TiO_2$ with Pt-$TiO_2$ interface showed higher selectivity in all methanol conversion regions. Plots of selectivity vs. conversion measured at 353 K were also shown in the Supplementary Fig. 9 and higher selectivity was observed for Pt nanowires/$TiO_2$ catalyst in all conversion regions at this temperature. Through this comparison of selectivity according to conversion, it can be seen that the metal-oxide interface plays a critical role in enhancing selectivity within the comparable methanol conversion region.

**Efficiency comparison of hot electron generation**. To elucidate the unique electronic structures of the nanoscale metal-oxide interface, we calculated the efficiency of the hot electrons produced by the chemical reaction using the chemicurrent and reactivity results. As the measured current of reaction-induced hot electron is linearly proportional to the reactivity[30,47], the chemicurrent generated by the catalytic methanol oxidation on the catalytic nanodiode can be described as Eq. (1)

$$I = \alpha q A N_{Pt} TOF \tag{1}$$

where $\alpha$ is the chemicurrent yield, $q$ is the elementary charge, $A$ is the active area of the Pt catalyst, $N_{Pt}$ is the number of Pt sites per $mm^2$, and TOF is the rate of methanol oxidation[10]. Therefore, we can obtain the chemicurrent yield, which is the probability of non-adiabatic electronic excitation forming one molecule of the product during a chemical reaction (i.e., the number of hot electrons captured per unit chemical reaction on the nanodiode surface). As shown in Fig. 2d, the chemicurrent yield, the efficiency of hot electron generation on the catalyst surface was compared for two cases. There was a reasonable chemicurrent density in Pt film/$TiO_2$ nanodiode, but it can be said that the reaction occurred completely on the Pt surface in the Pt film because the interfacial sites were not exposed to the methanol oxidation environment. In the Pt film/$TiO_2$, the thickness of the Pt film was 5 nm, making the effect of the substrate negligible (see Supplementary Note 3 for details). Selectivity was low in Pt film without Pt-$TiO_2$ interface (Fig. 2b, c), which resulted in a smaller hot electron excitation than in Pt nanowires/$TiO_2$ nanodiode. In addition, all the experimental results (e.g., TOF, chemicurrent density, and chemicurrent yield) were calibrated to the surface area of Pt. The number of Pt sites for nanowire arrays was calculated by geometrical assumptions (e.g., the Pt nanowire was considered to be rectangular and standing upright on a planar oxide support); therefore, the surface area effect of nanowires can

be excluded. As demonstrated by the previous results in the Pt film/$TiO_2$ nanodiode, there is a significant influence of selectivity to chemicurrent yield (i.e., as selectivity to methyl formate increased, the generation of hot electrons was enhanced); due to this fact, the chemicurrent yield of Pt nanowires/$TiO_2$ was higher than Pt film/$TiO_2$, owing to the higher selectivity of Pt nanowires/$TiO_2$ that have an exposed Pt-$TiO_2$ interface. We note that in our previous research, we found that hot electrons are generated much more in the reaction to methyl formate than $CO_2$[30] and it can be concluded that enhanced hot electron excitation obtained on Pt nanowires/$TiO_2$ indeed originates from their improved selectivity on nanoscale Pt-$TiO_2$ interfacial sites. Recently, we fabricated a Schottky nanodiode that formed a Pt/CoO interface and reported a significant increase in catalytic activity and hot electron excitation by hydrogen oxidation[48,49]. Thus, the improved partial oxidation selectivity when Pt nanowires were supported on $TiO_2$ can be attributed to the Pt-$TiO_2$ interfacial sites formed in the nanodiode, and owing to this increased selectivity, the efficiency of hot electron excitation was enhanced. Hence, the most striking finding of these results was that the Pt-$TiO_2$ interface could promote the partial oxidation reaction to form methyl formate and thereby enhance the hot electron generation on the catalyst. Furthermore, from the observation of change in the chemicurrent yield according to the partial pressure of methanol (i.e., according to product formation), we can also demonstrate that the efficiency of hot electron excitation was greatly affected by selectivity to methyl formate, not total activity (see Supplementary Note 4 and Supplementary Figs. 10 and 11 for details). In addition, for reliable comparison between the interface effects in multi-path reactions and those in one-path reactions, we carried out comparative experiments and examined the results of TOF and chemicurrent for the Pt nanowires/$TiO_2$ and the Pt film/$TiO_2$ in a hydrogen oxidation reaction whose product is only water. In the hydrogen oxidation reaction (i.e., hydrogen 4 Torr and oxygen to 760 Torr), the change in activity and the efficiency of hot electron excitation by the Pt-$TiO_2$ interface did not appear (Supplementary Note 5 and Supplementary Figs. 12 and 13), which indicates that selectivity is the key factor in influencing chemicurrent yield.

**Control experiments changing stack order and oxide support**. To further demonstrate the effect of the Pt-$TiO_2$ interface, we conducted further control experiments that changed the stack order or composition of supported nanowires. This involved fabricating $TiO_2$ nanowires/Pt inverse structure with $TiO_2$ nanowires deposited on Pt thin film and Pt nanowires/$SiO_2$ structure, and obtaining the selectivity and chemicurrent yield, as shown in Fig. 2e, f. We obtained and compared all the reaction data (e.g., TOF and selectivity) at each nanodiode by measuring the reaction rate at low methanol conversion (i.e., <10%) under a kinetically controlled regime (Supplementary Fig. 14). As shown in Supplementary Fig. 15a, we fabricated the inverse structure of $TiO_2$ nanowire arrays on the Pt film/$TiO_2$ diode with exactly the same size and period of nanowires with width of 20 nm (i.e., $TiO_2$ nanowires (width 20 nm)/Pt film/$TiO_2$ inverse catalytic nanodiode) to form the inverse $TiO_2$-Pt interface. The device was constructed, as shown in Supplementary Fig. 15b, and the Schottky barrier height was 0.88 eV, which was similar to the Pt film/$TiO_2$ and Pt nanowires/$TiO_2$. Under methanol 4 Torr and oxygen to 760 Torr, the steady-state chemicurrent was measured, and the stability of the device under reaction condition was confirmed by checking the unchanged electrical properties (Supplementary Fig. 15c, d). As the width of the $TiO_2$ nanowire arrays on Pt film was the same as the previous Pt nanowire arrays on the $TiO_2$ structure, the ratio of interface to metal site was also

the same for both structures (i.e., Pt nanowires (width 20 nm)/ TiO$_2$ and TiO$_2$ nanowires (width 20 nm)/Pt film/TiO$_2$). For this reason, the selectivity of the inverse structure was similar to those of the previous structure in which the Pt nanowires were on the TiO$_2$ with the same width, as shown in Fig. 2e. As the selectivity in Pt nanowires/TiO$_2$ and TiO$_2$ nanowires/Pt was almost the same, the chemicurrent yields in both systems were also similar (Fig. 2f). Therefore, whether the Pt nanowires were on TiO$_2$ or the TiO$_2$ nanowires were on the Pt film, the same amount of interface ratio could not affect the selectivity or hot electron generation; if the same ratio of metal-oxide interfacial sites formed, it can be seen that the stack order had little effect on the chemical reaction and hot electron excitation was hardly affected. This control experiment in depositing TiO$_2$ nanowires on the Pt film indicates that the nanoscale Pt-TiO$_2$ interface plays an important role in determining selectivity and hot electron generation in methanol oxidation.

Instead of the Pt-TiO$_2$ interface, the catalytic reaction experiment was carried out after depositing the Pt nanowire arrays on a non-reducible SiO$_2$ support known to have little metal-support interface effect[50,51]. In the structure where the Pt nanowires were placed on the SiO$_2$ support, the selectivity to produce methyl formate was not high, coming close to that of the Pt film/TiO$_2$ (Fig. 2e). Hence, the shape and size effect of the Pt nanowires could be excluded from the results of similar selectivity for Pt nanowires/SiO$_2$ and Pt film. Unlike on other oxide supports, previously reported results also showed similar activity to Pt nanowire arrays on SiO$_2$ supports and to both Pt film and single Pt(111) crystals[33]. Through this control experiment in which Pt nanowires were deposited on SiO$_2$, which is a non-reducible support, it was found that the Pt-TiO$_2$ interface has a greater effect on selectivity improvement than the Pt-SiO$_2$ interface. Pt nanowires may have exhibited different selectivity when deposited on SiO$_2$ and TiO$_2$ supports due to the difference in Lewis acidity between the two oxide supports[52].

To demonstrate the Lewis acid/base properties of these two oxide supports in a methanol oxidation reaction environment, we investigated the charge transfer from the adsorbate to the support surface when reactant methanol molecule was adsorbed on TiO$_2$ and SiO$_2$ support by using DFT calculation (see Supplementary Note 6 and Supplementary Figs. 16 and 17 for calculation details). As shown in Supplementary Fig. 17, when the methanol was adsorbed on two oxide surfaces, the electron density on the methanol basis was calculated (with a negative sign indicating a loss of electrons), yielding $-0.07$ when adsorbed on the TiO$_2$ surface and $+0.01$ when adsorbed on the SiO$_2$ surface. This result implies that when the adsorption of methanol occurs on the two oxide surfaces, electron transfer from methanol to surface occurs on the TiO$_2$ surface (i.e., Lewis acid) and electron transfer from surface to methanol occurs a little on the SiO$_2$ surface (i.e., Lewis base). Therefore, through this Bader charge analysis, it was found that the Lewis acidity of TiO$_2$ is stronger than that of the surface of SiO$_2$ when methanol oxidation reaction occurs. Therefore, when Pt nanowires were deposited on the TiO$_2$ support, which has a higher Lewis acidity than SiO$_2$, it showed higher partial oxidation selectivity. A previous study also indicated that the selectivity of methyl formate production in the methanol oxidation reaction increased as the Lewis acidity of the oxide support increased when the metal catalyst was introduced to the oxide support[53]. As a result, it has been confirmed that the Pt-SiO$_2$ interface had little metal-support interface effect compared to the Pt-TiO$_2$ interface, due to the different acid/base properties of SiO$_2$ and TiO$_2$. Hence, the increase in selectivity only when Pt nanowires were deposited on TiO$_2$, which was reducible support (i.e., strong Lewis acidity), rather than non-reducible SiO$_2$, lends

strong evidence to the claim that catalytic reactions occur mainly at nanoscale Pt-TiO$_2$ interface sites.

**Change of selectivity and efficiency of hot electron.** Because of a lack of definitive experimental evidence, the fundamental role and effect of metal-oxide interfacial sites are still debated. To prove this hypothesis, it is necessary to observe the change in selectivity according to the ratio of the Pt-TiO$_2$ interfacial sites. The change in selectivity as a function of the density of the metal-oxide interface provides evidence that the reaction primarily occurs at interfacial sites of metal-support when reducible oxides are used as supports for metal nanocatalysts[15,54]. Recently, we found a shift in catalytic activity when controlling the concentration of the metal-oxide interface under H$_2$ oxidation and CO oxidation, indicating that higher catalytic activity was observed when more metal-oxide interfaces were created[55,56]. Moreover, a previous study also reported that higher butanol production selectivity was achieved by generating more metal-oxide interfaces, as the density of Pt nanoparticles on oxide support increases by changing the Langmuir–Blodgett surface pressure[57]. In addition, when Pt, Pd, and Ni nanocatalysts were supported by oxide, there was a previous study in which the CO oxidation reactivity increased as the size of the nanocatalysts decreased, since the ratio of interfacial sites (i.e., perimeter and corner atoms) increased as the particle size decreased[17]. Therefore, to further investigate the effect of the metal-oxide interface ratio, we fabricated a nanodiode based on Pt nanowire arrays with width of 15 nm; the ratio of interface to metal sites of these nanowire arrays was higher than the earlier Pt nanowire arrays with width of 20 nm. In other words, a higher ratio of Pt-TiO$_2$ interface to metal sites could be modeled by decreasing the width of Pt nanowires. Concurrently, we fabricated the Pt nanowire arrays by increasing the width to 50 nm and depositing it on the TiO$_2$ support to produce a catalytic nanodevice. To further observe the selectivity and hot electron excitation changes according to the width control of Pt nanowires, both the selectivity and chemicurrent yield were measured as the width of the nanowires was changed (see Supplementary Note 7 and Supplementary Figs. 18 and 21 for details).

Figure 3a–c represent the SEM image of the fabricated nanowire arrays of Pt nanowires (width 15 nm)/TiO$_2$, Pt nanowires (width 20 nm)/TiO$_2$, and Pt nanowires (width 50 nm)/TiO$_2$, respectively. The selectivity and quantity of hot electron generation from methanol 4 Torr and oxygen to 760 Torr in each nanodiode were compared according to the width of Pt nanowires by adding the results from the fabricated devices (Fig. 3d, e). All the current density measured under methanol oxidation and non-reactive conditions is shown in Supplementary Fig. 22. As shown in Fig. 3d, compared to Pt film, selectivity increased when Pt nanowires were deposited on TiO$_2$ support and it was confirmed that selectivity was enhanced as the width of the Pt nanowires decreased. As the width of Pt nanowires on TiO$_2$ was reduced from 50 nm to 15 nm, the ratio of Pt-TiO$_2$ interface to metal increased, which enhanced selectivity to methyl formate formation, owing to the increased ratio of interfacial sites. In addition, when comparing selectivity according to methanol conversion, the selectivity trend according to the width of nanowires was clearly shown in all methanol conversion regions (Supplementary Fig. 23).

As shown in Fig. 3e, when comparing the chemicurrent yield detected in catalytic nanodevices as a function of the width of Pt nanowires, it can be seen that the chemicurrent yield increased (i.e., higher efficiency for reaction-induced hot electron excitation) as the width of nanowires decreased, owing to the enhanced

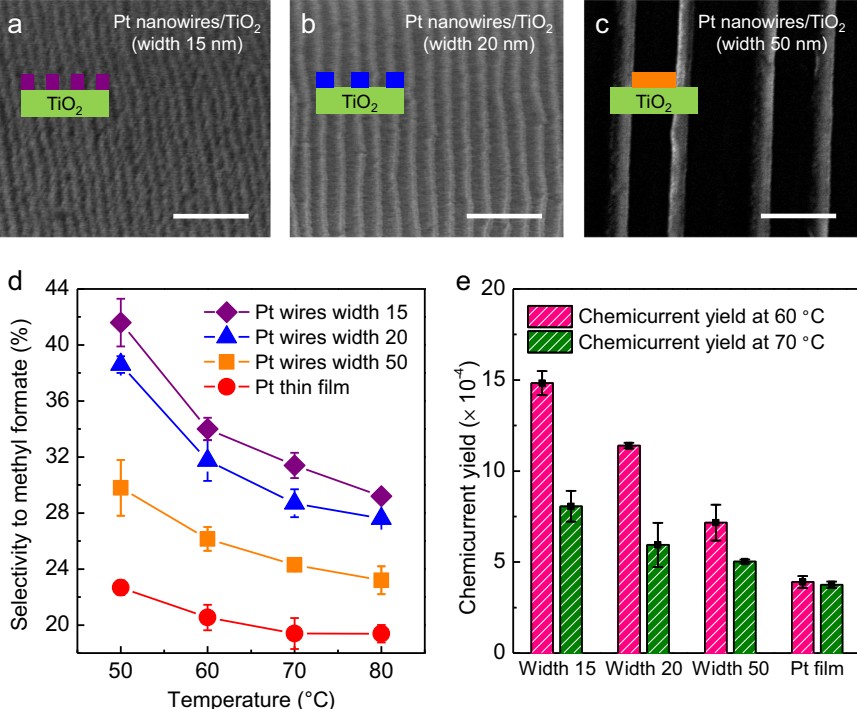

**Fig. 3 Selectivity and chemicurrent yield on nanowire arrays with different widths.** SEM images of **a** Pt nanowires/$TiO_2$ with a width of 15 nm, **b** Pt nanowires/$TiO_2$ with a width of 20 nm, and **c** Pt nanowires/$TiO_2$ with a width of 50 nm. Scale bars are 200 nm. **d** Selectivity to methyl formate under methanol oxidation on the Schottky nanodiodes at different temperatures with different widths. **e** Chemicurrent yield for methanol oxidation on the catalytic nanodiodes of width 15 (Pt nanowires/$TiO_2$ with a width of 15 nm), width 20 (Pt nanowires/$TiO_2$ with a width of 20 nm), width 50 (Pt nanowires/ $TiO_2$ with a width of 50 nm), and Pt film (Pt film/$TiO_2$) measured both at 333 and 343 K. All results were obtained under methanol 4 Torr and $O_2$ to 760 Torr.

selectivity by the increased ratio of interfacial sites. Therefore, we can conclude that as the width of the Pt nanowires decreased, higher selectivity to methyl formate was observed due to the increased ratio of Pt-$TiO_2$ interface, thereby enhancing hot electron excitation in catalytic nanodiodes. In all nanodiodes, the Schottky barrier height was obtained by fitting each $I–V$ curve to thermionic emission theory, as shown in Supplementary Table 3. As mentioned earlier, this Schottky barrier acts as a filter for hot electron transfer, and only energetic electrons whose excess chemical energy by energy conversion is higher than the Schottky barrier height can be detected on this metal–semiconductor nanodiode. Here, despite the higher Schottky barrier height in the Pt nanowires/$TiO_2$ nanodiode with width of 15 nm (i.e., ~1.02 and 0.85 eV for Pt nanowires (width 15 nm)/$TiO_2$ and Pt nanowires (width 20 nm)/$TiO_2$, respectively), the higher hot electron detection indicates that the enhanced selectivity owing to the increased Pt-$TiO_2$ interface concentration has a significant effect on hot electron excitation, neglecting the decrease in hot electron transfer due to the increased barrier. Therefore, as the structure exhibiting smaller width has a higher ratio of metal-oxide interface to metal sites, it exhibits higher partial oxidation selectivity to form methyl formate, thereby increasing reaction-induced hot electron excitation in methanol oxidation. The width dependence of selectivity of these $TiO_2$ supported nanowires indicates that the selective partial oxidation reaction primarily occurs at the metal-oxide interface. Thus, the well-controlled width of oxide supported nanowires through our lithography technique can easily tune the ratio of metal-oxide interface, which changed the catalytic reaction significantly.

**DFT calculations for two possible pathways**. To gain further insight into the enhanced selectivity of methyl formate on

Pt-$TiO_2$ interface compared to Pt film, we performed DFT calculations on the Pt nanorod model on $TiO_2$(110). Two possible reaction pathways toward $CO_2$ and methyl formate formation are laid out in Supplementary Fig. 24, and detailed mechanisms, including all elementary steps and associated energetics are presented in Supplementary Table 4, following previously suggested mechanisms[30,58]. First, as shown in Supplementary Fig. 24, methanol is oxidized to formaldehyde, which can further oxidize to form $CO_2$ or undergo C–C coupling to form methyl formate. As the reaction of the formaldehyde intermediate determines which product is produced between methyl formate and $CO_2$[26], we considered the reaction channel from formaldehyde in two catalysts to compare the selectivity in the methanol oxidation reaction. The free energetics from formaldehyde to both products are presented in Fig. 4a, b. To initiate the reaction, methanol preferred to adsorb on the Ti top site of $TiO_2$, whereas oxygen preferred the Pt sites in the present Pt-$TiO_2$ interfacial structure. Thus, the reaction would mainly take place by transferring the protons from adsorbates on the Ti sites to oxygen (or OH) on Pt. However, once *$CH_2O$ loses its proton to *OH (*$CH_2O +$ *OH $\rightarrow$ *HCO + *$H_2O$), *HCO prefers and moves to the Pt sites. Consequently, the final reaction CO + O $\rightarrow CO_2$ occurs above the Pt sites.

As the selectivity is determined by the activation barrier in the reaction determining steps (RDSs), we calculated the reaction energy barriers for $CO_2$ and methyl formate on the Pt nanorods on $TiO_2$(110) modeling Pt nanowires supported on $TiO_2$. The Pt-$TiO_2$ energetics were then compared to our previous calculation on a three-atomic-layer of Pt(111) surface, representing the reaction on the Pt film surface (Supplementary Table 5)[30]. As shown in Fig. 4a, on Pt-$TiO_2$(110), the largest barrier step for the methyl formate formation was the last dehydration step

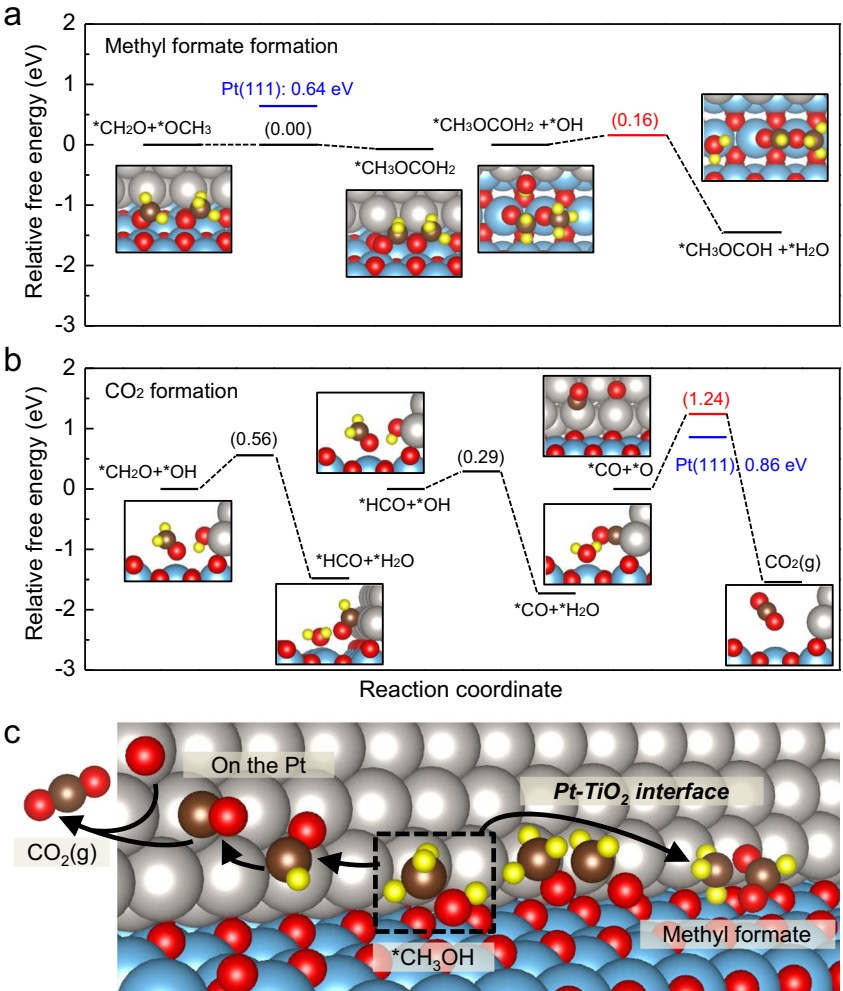

**Fig. 4 Calculated activation barrier for Pt nanorod on TiO₂(110) compared with Pt(111).** Optimized free-energy profiles for **a** methyl formate and **b** the $CO_2$ formation path with reaction barriers ($\Delta G‡$). Barriers with red and blue indicate the RDS of each path for Pt nanorod structure on TiO₂(110) and Pt (111), respectively. **c** Schematic drawing showing formation of $CO_2$ and methyl formate in methanol oxidation on Pt nanorod/TiO₂(110). Species marked with an asterisk (*) are adsorbed on the surface. The gray, blue, brown, red, and yellow balls indicate Pt, Ti, C, O, and H, respectively.

($*CH_3OCOH_2 + *OH \rightarrow *CH_3OCOH + *H_2O$) with a low barrier of 0.16 eV. In the case of Pt(111), the RDS (i.e., the C–C coupling step) was different, with a much higher barrier of 0.64 eV. The altered RDS was attributed to the different adsorption configuration of $CH_2O$ that is needed for C–C coupling and has changed in the respective sites, as shown in Supplementary Fig. 25. In other words, on Pt(111), C and O are simultaneously adsorbed on the surface, requiring the rupture of the metal-C bonding for C-C coupling, whereas on Pt-TiO₂(110) only O is bonded to the surface with an easy rotation of the methyl group for the coupling reaction.

The RDS in the $CO_2$ formation path on both Pt-TiO₂(110) and Pt(111) is the last C–O coupling step ($CO + O \rightarrow CO_2$), but the activation barrier was much larger on Pt-TiO₂(110) (1.24 eV) than on Pt(111) (0.86 eV). This intriguing result could be attributed to a change in the reaction mechanism; on Pt(111), the reaction occurs via the Langmuir–Hinshelwood mechanism ($*CO + *O \rightarrow CO_{2(g)}$), whereas on the interfacial Pt sites of the Pt-TiO₂(110), the reaction follows the Eley–Rideal mechanism ($CO_{(g)} + *O \rightarrow CO_{2(g)}$) where CO desorbs first and then forms a new bond with adsorbed O. The binding energy of $CO_2$ on the TiO₂ surface was calculated, because the lower production of $CO_2$ may be due to the adsorption of $CO_2$ on TiO₂. As shown in Supplementary Fig. 26, the binding energy of $CO_2$ on the TiO₂ surface was −0.21 eV, which is an energy level

that can be sufficiently desorbed even at room temperature[59]; thus, the reason for the adsorption of $CO_2$ on the TiO₂ cannot be the low production of $CO_2$. In addition, in the theoretical calculation on the Pt nanorod/TiO₂(110) model, HCO, which is the intermediate before $CO_2$ is generated, prefers adsorption to the Pt surface and $CO_2$ generation occurs at the Pt site; thus, $CO_2$ adsorption in TiO₂ can also be excluded. Furthermore, as mentioned above, since the reaction mechanism in which $CO_2$ was generated by methanol oxidation was revealed by DFT calculation as an Eley–Rideal mechanism ($CO_{(g)} + *O \rightarrow CO_{2(g)}$) in the Pt nanorod/TiO₂(110) model, there was no state in which $CO_2$ was adsorbed during the methanol oxidation reaction. Thus, the experimentally observed enhanced selectivity toward methyl formate over $CO_2$ on the Pt nanorod on TiO₂(110) compared to thin Pt film can be understood by the much lower activation barrier to form methyl formate (0.64 for Pt(111) $\rightarrow$ 0.16 eV for Pt-TiO₂(110)), as well as the increased activation barrier to form $CO_2$ (0.86 for Pt(111) $\rightarrow$ 1.24 eV for Pt-TiO₂(110)). The pathways in which $CO_2$ and methyl formate were generated in Pt nanorod/TiO₂(110) investigated by theoretical simulation are shown in Fig. 4c. Here, $CO_2$ was generated on the Pt surface and methyl formate formed by reacting the intermediate present on TiO₂ with OH present on Pt, and the importance of the Pt-TiO₂ interface to improve the selectivity of methyl formate production could be theoretically proven. Hence, we can prove the

experimental results of the reduced $CO_2$ production and increased methyl formate production in Pt nanowires/$TiO_2$ compared to Pt film/$TiO_2$ (Fig. 2b, c). We note that a similar effect, changing activation barriers when the interfacial sites formed, was also demonstrated in the previous combined theory and experimental studies for Pt-NiO and Pt-CoO interfaces[49,60]. Therefore, all these results can point to the conclusion of selectivity enhancement on metal-oxide interfacial sites in Pt nanowires/$TiO_2$, as these theoretical calculation results demonstrate the increased barrier for $CO_2$ generation and the decreased barrier for methyl formate production. As hot electrons generated by non-adiabatic electronic excitation by exothermic catalytic reaction are excited states, it is challenging to directly calculate the transfer of excited hot electrons through DFT calculation and compare them in these two systems. However, we note that our earlier DFT calculations[30] show the relationship between the generation of methyl formate and hot electron generation. In this study, it was found that the selectivity of methyl formate formation was enhanced when the Pt/$TiO_2$ interface was formed, which was due to the difference in the activation barrier confirmed by theoretical calculations. Our quantitative comparison between the selectivity of methyl formate and $CO_2$ on Pt-$TiO_2$ and that of Pt(111) via DFT calculations, the enhanced hot electron generation on Pt-$TiO_2$ interface can be rationalized. These are important findings, because these works are the first visualization of the unique electronic property of the metal-oxide perimeter in oxide supported-metal nanowires obtained by real-time measurement of excited hot electrons from the exothermic surface chemical reaction.

## Discussion

In this study, we have investigated the metal-oxide interface effect on partial oxidation selectivity using a newly designed Schottky nanodiode composed of two-dimensional Pt nanowire arrays deposited on $TiO_2$, and the changed selectivity influenced the magnitude of reaction-induced hot electron flow excited by methanol oxidation. Compared to Pt films on $TiO_2$ where the interface was not exposed to the reaction environment, the Pt-$TiO_2$ interface was exposed to the gaseous environment when the Pt nanowire arrays were placed on $TiO_2$. The effect of the metal-oxide interface can be identified by comparing the selectivity in these two structures. We observed that the formation of the nanoscale Pt-$TiO_2$ interface showed that selectivity to methyl formate formation was much larger in Pt nanowire arrays than in Pt films. As the efficiency of hot electron excitation by exothermic catalytic reaction was attributed to selectivity, the chemicurrent yield showed much higher in Pt nanowire arrays than on Pt film owing to enhanced selectivity, confirming the role of metal-oxide interfacial sites in enhancing selectivity and reaction-induced electron transfer. Calculation of the activation barrier by DFT calculation in two models of Pt nanorod/$TiO_2$(110) and Pt(111) revealed barrier differences due to different molecular adsorption orientations and reaction mechanisms in the two models, which indicates improved selectivity at the Pt-$TiO_2$ interface. This study is the first to directly observe the reaction-induced electronic excitation at the nanoscale metal-oxide interface, which was made possible by the real-time detection of hot electrons excited by catalytic reaction in the newly designed Pt nanowires/$TiO_2$ system. Thus, this technique for using the catalytic nanodiodes provides a powerful and highly sensitive tool for studying the processes of charge transfer at the metal-oxide interface excited by surface chemical reaction. These studies demonstrate that the presence of nanoscale metal-oxide interfaces alters selectivity, thereby influencing reaction-induced electron transfer in heterogeneous catalysts. Tuning the materials and architectures of

nanowires catalysts gives rise to the important control mechanism of chemical reaction and hot electron generation.

## Methods

**Fabrication of master molds for nanowire transfer printing**. Cylinder forming polystyrene-block-poly(dimethylsiloxane) (PS-$b$-PDMS) block copolymers with molecular weights (MWs) of 36 kg/mol (SD36) and 48 kg/mol (SD48), which form 15- and 20 nm-wide lines, and hydroxyl-terminated PDMS brush polymer with a MW of 5 kg/mol were purchased from Polymer Source, Inc. (Canada). Surface-patterned Si substrates with a width of 1 μm and a period of 1.2 μm were fabricated using KrF photolithography followed by reactive ion etching and used as guiding substrates for the self-assembly of block copolymers. The hydroxyl-terminated PDMS solution dissolved in heptane solvent with a 2 wt% was spin-coated on the pre-patterned Si substrate. After spin casting, the sample was thermal annealed at 150 °C and then washed with heptane, to remove unattached polymer residues. SD36 and SD48 block copolymers were dissolved in a mixed solvent of toluene, heptane, and PGMEA (1 : 1 : 1 by volume) with a 0.6–0.8 wt% polymer solution. The SD36 and SD48 polymer solutions were spin-cast on the prepared substrates, respectively, and solvent-annealed with toluene vapor for 6–10 h in the chamber at room temperature to form well-ordered line/space patterns. After the annealing process, the samples were etched with $CF_4$ plasma followed by $O_2$ plasma treatment to obtain oxidized line/space PDMS nanostructures. To form Pt nanowires with 50 nm width and 200 nm pitch, the master mold was fabricated using KrF photolithography followed by reactive ion etching process.

**Fabrication of n-type $TiO_2$ support for catalytic nanodiodes**. A 200 nm titanium film was deposited by electron beam evaporation with a titanium target using a patterned aluminum shadow mask ($4 \times 6$ mm$^2$) on a 4 inch 500 nm $SiO_2$-covered $p$-type Si (100) wafer. After its deposition, the titanium film was thermally oxidized in air at 470 °C for 2.5 h, forming an $n$-type $TiO_2$ film; the sheet resistance was checked during annealing. The electrode with a 50 nm Ti and a 150 nm Au thin film was deposited sequentially through a second shadow mask ($5 \times 5$ mm$^2$) on both sides by electron beam evaporation. The Ti layer was deposited to form ohmic contact with the $n$-type $TiO_2$ surface and the Au layer created the two Ohmic electrodes for the catalytic nanodiodes. At room temperature, all the metal films were deposited with 10 kV of DC voltage in a high-vacuum chamber with a base pressure of $2 \times 10^{-8}$ Torr, which was evacuated using rotary and turbomolecular pumps. The deposition rate of metal was 0.2 Å s$^{-1}$ until a thin film of 5 nm was deposited for adhesion on the surface and the deposition rate was then gradually increased to 1.0 Å s$^{-1}$ until reaching the final thickness of the metal film.

**Solvent-assisted nanotransfer printing of Pt nanowire arrays on $TiO_2$ support**. Poly(methyl methacrylate) (PMMA; MW = 100 kg/mol) was purchased from Sigma Aldrich, Inc. and dissolved in a mixed solvent of toluene, acetone, and heptane with four wt% (4.5 : 4.5 : 1 by volume). The surface of the master molds was treated with a hydroxyl-terminated PDMS brush to enable the release of the polymer replica film from the master molds. After surface treatment, a PMMA polymer solution was spin-coated onto the master molds. A polyimide (PI) adhesive film (3M, Inc.) was then attached to the top surface of PMMA and smoothly detached from the master mold with the inverted surface topography. Pt nanowires were fabricated by obliquely angled deposition using an e-beam evaporator. The fabricated Pt nanowires/PMMA/PI adhesive film was exposed to a mixed solvent vapor of acetone and heptane (1 : 1 by volume) at 55 °C for 20–30 s then immediately brought into contact with the $TiO_2$ surface of a nanodiode. After the transfer process, the PMMA replica was removed by dipping it into the toluene solvent baths. In addition, to facilitate the movement of hot electrons generated at the interface between Pt nanowires and $TiO_2$, catalytic-inert Au nanowires with a width of 200 nm and a period of 1.2 μm were also transferred to the fabricated nanodiodes in a mesh type by sequential printing using the same method.

**Measurement of reaction rates and chemicurrent**. Catalytic methanol oxidation reactions were performed in an ultrahigh vacuum chamber (1 L) with a base pressure of $1 \times 10^{-8}$ Torr, which was evacuated using rotary and turbomolecular pumps. After isolating with a gate valve, the catalytic reaction was carried out in a batch reactor system filled with a mixture of 1, 2, or 4 Torr of methanol and $O_2$ to atmospheric pressure (760 Torr) at room temperature. To inject the methanol vapor into the reaction chamber, the methanol reactant was purified by three cycles of freeze–pump–thawing. All the reactant gas mixtures were recirculated through the reaction line using a Metal Bellows circulation pump at a rate of 2 L/min. After 0.5 h of recirculation to reach equilibrium at room temperature, a gas chromatograph (DS 6200) equipped with a thermal conductivity detector, for detection of $CO_2$ product, and a flame ionization detector, for detection of methanol reactant and methyl formate product, were used to separate the gases for analysis. The reaction rates were reported as TOF in units of product molecules produced per Pt surface site pert second, and the number of Pt sites for nanowire arrays was calculated by geometrical assumptions. The Pt nanowire was assumed to be rectangular and standing upright on a planar oxide support similar to previous

study[33]. The signals of electric current for the chemicurrent generated on the catalytic Schottky nanodiode were obtained using a Keithley Instrumentation 2400 SourceMeter at open circuit. Two gold wires were connected to make contact with the two electrical contact pads on either side (Au/Ti layer). The electric current flowing across the Schottky junction was then measured at reactive conditions (1–4 Torr of methanol and $O_2$ to equal 760 Torr) and at non-reactive conditions (760 Torr of pure $O_2$).

**Characterization of fabricated catalytic nanodiodes.** The crystalline phases of the Pt nanowire arrays were revealed using XRD patterns (Rigaku D/MAX-2500) taken at a $2\theta$ scan range of 20–90°, scan speed of 4° min$^{-1}$, and step size of 0.01 Å using Cu Kα radiation. The chemical oxidation states of the Pt nanowires (Pt 4 f) before and after catalytic reaction were identified using XPS (Thermo VG Scientific Sigma Probe spectrometer with an X-ray source of Al K-alpha (1486.3 eV)) with an energy resolution of 0.5 eV full width at half maximum under ultrahigh vacuum conditions with a base pressure of $10^{-10}$ Torr. Field-emission SEM (Magellan 400) and TEM (FEI Titan G2 Cubed 60-300) were used to observe the morphological structures of the Pt nanowires. I–V characteristics of the Pt nanowires/TiO$_2$ catalytic nanodiodes were obtained using a Keithley Instrumentation 2400 Source-Meter under various gas conditions (mixtures of 1–4 Torr of methanol with $O_2$ to equal 760 Torr). By fitting the obtained I–V characteristics to the thermionic emission equation, a series resistance, Schottky barrier height, and ideality factor could be obtained.

**Simulation methods.** DFT calculations were carried out using the VASP code with the Perdew–Burke–Ernzerhof functional[61,62]. The projector augmented wave method was used to describe the potentials of the atoms[63]. Geometries were optimized until the force on each atom was less than 0.05 eV/Å with a cutoff energy of 400 eV. As shown in Supplementary Fig. 16, to simulate the Pt-TiO$_2$ structure, a $(4 \times 3)$ unit cell was used for rutile TiO$_2$(110), which contains two layers of TiO$_2$, and a Pt nanorod three layers high and three atoms wide is bonded on top of the TiO$_2$(110) surface[21]. The bottom layer of TiO$_2$ was fixed to their bulk position and all of the Pt atoms are allowed to relax in the Z-direction. K-points were sampled using a $2 \times 1 \times 1$ Monkhorst–Pack grid. To model the Pt film, a three-atomic-layer $(3 \times 3)$ of Pt(111) surface unit cell with the bottom layer fixed was used. Spacing of more than 15 Å in the z-direction was applied. The Gibbs free energies were calculated using $\Delta G = \Delta E_{ads} + \Delta E_{ZPE} - T\Delta S$ ($T = 300$ K), where $\Delta E_{ads}$, $\Delta E_{ZPE}$, and $\Delta S$ represent the changes in adsorption energy, zero-point energy, and entropy, respectively. The DFT + U corrections were applied to the metal d states to correct the on-site Coulomb interaction, and the U-value was set at 4.0 eV[64]. The climbing image nudged elastic band method was employed to calculate the activation barriers[65]. The Bader charge analysis was performed to estimate the amount of charge transfer after methanol adsorption on Pt-TiO$_2$ and Pt-SiO$_2$[66]. Negative sign indicates the loss of electrons.

## Data availability

The data that support the findings of this study are available from the corresponding author upon reasonable request.

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

## Acknowledgements

This work was supported by the Institute for Basic Science (IBS) [IBS-R004]. Y.J. acknowledges support through the National Research Foundation of Korea from the Korean Government (NRF-2019M3D3A1A01069099 & 2016M3D1A1021147).

## Author contributions

S.W.L. and J.Y.P. planned and designed the experiments. S.W.L. performed the central measurements and analysis. S.W.L. and H.L. fabricated the nanodevices and carried out the chemicurrent and catalytic activity measurements. W.P. and Y.J. performed the DFT calculations. J.K., G.R.L., and Y.S.J. fabricated the nanowire-based diode. All authors discussed the results and contributed to the manuscript.

## Competing interests

The authors declare no competing interests.
