## [Peer Review File · Nature Communications]

Reviewers' comments:

Reviewer #1 (Remarks to the Author):

This manuscript describes the preparation of a series of Pt nanowire arrays where the purpose of the experiments is to investigate the role of the metal-oxide interface in affecting selectivity and to examine the potential role of hot electrons. This manuscript is potentially of interest, but it is very poorly organized and difficult to read. Most of the manuscript asserts results without providing enough experimental data. Results and discussion are completely mixed throughout the manuscript, and within individual paragraphs making it very difficult to determine what is new, what has been previously published, what is data, and what is speculation. There do not appear to be sufficient control experiments to determine the assertions.

In particular, selectivity must be measured at comparable conversions. The selectivity differences are relatively small, and the conversions are unknown. While the selectivity differences might be due to the interface, it is not clear that this is due to the hot electrons – the different acid/base properties of silica and titania could also be driving this difference.

The data are probably interesting, but the manuscript should be completely re-written so that it is clear, coherent, and accessible to a broader audience if it is going to appear in Nature Communications.

Some details:

1. The manuscript's use of the SMSI is incorrect. SMSI refers to highly mobile support that can move onto the metal; this is typically associated with poisoning of the metal surface, not activating the metal.
2. While the manuscript presents a nice cartoon showing catalysis occurring at the metal-oxide interface, the experimental evidence for it was less convincing. Simply changing the pitch doesn't change the ratio of interface to metal sites – the width of the wires needs to be changed.
3. It is not clear what the Ptfilm/TiO₂ is or how it differs from the nanowire arrays. This is critical as this provides a possible control where there is little or no exposed interface. The film shows reasonable electrical activity, which suggests that the higher activity of the nanowires is only a Pt surface area effect and therefore that the reaction occurs on the Pt (and not at the interface)

Reviewer #2 (Remarks to the Author):

Recommendation: Publish in Nature Com after minor revision

Comment to the Author:

In this manuscript, the author construct Pt nanowires/TiO₂ catalytic nanodiode to explore the properties of metal-oxide interface. The relation between catalytic activity and selectivity was related to the hot electron generation. DFT was also carried out to confirm the critical role of Pt-TiO₂ interface. We highly appreciate the new method to reveal the relationship between catalytic performance and hot electron generation. However, there was some cracks still present in this manuscript. If the author successfully address these problems, we will be glad to accept this paper. Thus, I recommend a minor revision, and list my comments as follows:

1. What is the relationship between the hot electron generation and the valence of Pt? where the hot electron generated from? Oxygen vacancies or metal Pt surface? If the interfacial oxygen vacancy promote the generation of hot electrons?
2. The reference is not up to date; And in order to help the author to deeply understand the interfacial synergistic mechanism, the author should read and cite the following references: J. Am. Chem. Soc. 2018, 140, 11241. ACS Catal. 2017, 7, 7600. ACS Catal. 2019, 9, 2707.

Reply to Reviewer 1's report

Referee comment #0: *This manuscript describes the preparation of a series of Pt nanowire arrays where the purpose of the experiments is to investigate the role of the metal-oxide interface in affecting selectivity and to examine the potential role of hot electrons. This manuscript is potentially of interest, but it is very poorly organized and difficult to read. Most of the manuscript asserts results without providing enough experimental data. Results and discussion are completely mixed throughout the manuscript, and within individual paragraphs making it very difficult to determine what is new, what has been previously published, what is data, and what is speculation. There do not appear to be sufficient control experiments to determine the assertions. In particular, selectivity must be measured at comparable conversions. The selectivity differences are relatively small, and the conversions are unknown. While the selectivity differences might be due to the interface, it is not clear that this is due to the hot electrons – the different acid/base properties of silica and titania could also be driving this difference. The data are probably interesting, but the manuscript should be completely re-written so that it is clear, coherent, and accessible to a broader audience if it is going to appear in Nature Communications.*

Reply #0: We thank the reviewer for their interest and valuable comments. Here, we modified the manuscript as a whole to explain the experimental results, because of the comments regarding a lack of composition and readability of manuscript, as mentioned below:

“...This manuscript is potentially of interest, but it is very poorly organized and difficult to read... Results and discussion are completely mixed throughout the manuscript, and within individual paragraphs making it very difficult to determine what is new, what has been previously published, what is data, and what is speculation...”.

Reflecting what the reviewer pointed out, we changed the overall organization of the manuscript and modified a lot for a better readable paper. In each paragraph we have added a description of our results compared to previously published studies, and our assertions are explained more clearly using experimental data. Unlike previous versions where results and discussions were mixed, they are clearly distinguished and revised. To support the points we were trying to claim, we did additional experiments and data interpretation pointed out by the reviewer, and changed the composition of the entire manuscript so that we completely re-write the manuscript. I hope the reviewer will look at the revised manuscript as a whole. Detailed responses to the reviewer's comments are given below.

We added the conversion results measured in a batch reactor for precise selectivity comparison to address the reviewer’s comments below:

“...In particular, selectivity must be measured at comparable conversions. The selectivity differences are relatively small, and the conversions are unknown.”

As shown in the additional results below (Fig. R1), it can be seen that the turnover number of products and methanol conversion increased linearly over time because the catalytic reaction was performed in a batch reactor system. Turnover frequency (TOF) representing catalytic activity was calculated from the slope of the turnover number with reaction time while monitoring methanol conversion. We obtained all the reaction data (e.g., TOF and selectivity) by measuring the reaction rate at low methanol conversion (i.e., less than 10 %) under a kinetically controlled regime. Therefore, we can compare our selectivity with comparable conversion. In response to comment from the reviewer, we added these conversion results (see Supplementary Fig. 18 or Fig. R1) and descriptions to the revised manuscript and supplementary information.

Fig. R1 | TON and methanol conversion with reaction time measured under 4 Torr of methanol and 760 Torr of O₂ at 343 K for **a.** Pt film/TiO₂, **b.** Pt nanowires (width 20 nm)/TiO₂, **c.** Pt nanowires (width 20 nm)/SiO₂, **d.** Pt nanowires (width 50 nm)/TiO₂ catalysts. All TOFs and selectivity were compared in a region with a methanol conversion of less than 10%.

Furthermore, we once again emphasized the reason why hot electron generation increased in the interface, and also it was further explained that the difference in selectivity when Pt nanowires were deposited on SiO₂ and TiO₂ (Fig. R2) was due to different Lewis acidity of support following the reviewer's comments, as mentioned below:

“...While the selectivity differences might be due to the interface, it is not clear that this is due to the hot electrons – the different acid/base properties of silica and titania could also be driving this difference.”

Hot electrons were generated by energy conversion when exothermic chemical reactions occur in Pt catalyst, and we detected them in real time using catalytic nanodiode (Figs. 2a and 2b). According to our previous study on Pt film/TiO₂ catalytic nanodiode, we observed experimentally that the production of hot electrons is boosted in the reaction pathway to methyl formate than CO₂ because the exothermic energy of the reaction to methyl formate is higher, which was also verified through theoretical calculation (Lee, S. W. *et al. ACS Catal.* **9**, 8424-8432 (2019)). When Pt nanowires were deposited on TiO₂ support instead of Pt film where the Pt-TiO₂ interface was not exposed, and catalytic nanodevices were fabricated, selectivity increased due to the nanoscale interface on Pt nanowires-TiO₂, as pointed out by the reviewer. When the Pt-TiO₂ interface was formed, the selectivity increase of methyl formate production was also verified by the change of the activation barrier through our DFT calculation, as shown in Fig. 4. As just mentioned, since there is a correlation between selectivity of methyl formate production and hot electron generation on the catalysts, it can be explained that chemiurrent yield (*i.e.*, efficiency for hot electron generation) increased in Pt nanowires/TiO₂ nanodiode with increased selectivity than in Pt film/TiO₂ nanodiode.

Instead of the Pt-TiO₂ interface, we performed the catalytic reaction by depositing Pt nanowires on SiO₂, a non-reducible support. As shown in the results below (Fig. R2), the selectivity in Pt nanowires (width 20 nm)/SiO₂ was not high, and it was almost similar to the value in Pt film/TiO₂ whose Pt-TiO₂ interface was not exposed to the environment of the gas-phase chemical reaction. Unlike on other reducible supports, previous studies conducted in Somorjai's group showed similar activity to Pt film or Pt(111) single crystals when Pt nanowires were deposited on non-reducible SiO₂ support (Contreras, A. *et al. Cat. Lett.* **111**, 5-13 (2006)). In addition, as the reviewer mentioned, it might be due to the difference in Lewis acidity between the two supports that Pt nanowires exhibited different selectivity when deposited on SiO₂ and TiO₂ supports (Hanukovich, S. *et al. ACS Catal.* **9**, 3537-3550 (2019)). Therefore, when Pt nanowires were deposited on TiO₂ support which is known to have higher Lewis acidity than SiO₂ (Baker, L. R. *et al. J. Am. Chem. Soc.* **134**, 14208-14216 (2012)), it showed high selectivity (as shown in Supplementary Fig. 17a). As a result, it has been confirmed that the Pt-SiO₂ interface had little metal-support interface effect compared to the Pt-TiO₂ interface due to the different acid/base properties of SiO₂ and TiO₂. Thus, the increase in selectivity only when Pt nanowires were deposited on TiO₂, which was reducible support (*i.e.*, strong Lewis acidity), rather than non-reducible SiO₂, can be strong evidence of

the claim that catalytic reactions occur mainly at nanoscale Pt-TiO₂ interface sites. We are so grateful that the reviewer pointed out what we were overlooking, and we added all of these results (Supplementary Fig. 17a or Fig. R2) and explanations in the revised manuscript and supplementary information, thus allowing this manuscript to clearly deliver the main message of our research work.

Fig. R2 | Selectivity to methyl formate under methanol oxidation on the Schottky nanodiodes at different temperatures with different composition or stack order of nanowires.

Referee comment #1. *The manuscript's use of the SMSI is incorrect. SMSI refers to highly mobile support that can move onto the metal; this is typically associated with poisoning of the metal surface, not activating the metal.*

Reply #1: We thank the reviewer for this meticulous comment regarding the meaning of strong metal–support interaction (SMSI). As pointed out by the reviewer, encapsulation of metal nanoparticles by reducible oxide-support overlayers, (*i.e.*, the migration of oxide support onto the active metal surface) is the classic mechanism by which oxide supports can affect the catalytic behavior of metal nanoparticles which was discovered by Tauster *et al.*, as SMSI (Tauster, S. J. *et al. J. Am. Chem. Soc.* **100**, 170-175 (1978)). Matsubu *et al.*, also showed that adsorbate-functionalized encapsulation of metal nanoparticles by the reducible support (*e.g.*, TiO₂, Nb₂O₅) can influence the selectivity of CO₂ hydrogenation in reducible oxide-supported heterogeneous Rh catalysts (*i.e.*, adsorbate-mediated SMSI effect), and these results were proved by observing changes in pre-treatment through *in-situ* diffuse-reflectance infrared Fourier transform spectroscopy and *in-situ* scanning transmission electron microscopy (Matsubu, J. C. *et al. Nat. Chem.* **9**, 120-127 (2017)).

In addition, charge transfer at the metal-support interface could change the electronic properties of the metal nanoparticles influencing their adsorption behavior. This

electronic perturbations was found recently by Rodriguez and co-workers, in which Pt nanoparticles deposited on CeO₂ support increased the catalytic activity of water-gas shift reaction by enhanced dissociation of O-H bonds (Bruix, A. *et al. J. Am. Chem. Soc.* **134**, 8968-8974 (2012)). This electronic perturbations by charge transfer between metal and support on interface was termed as ‘electronic metal-support interaction’ (Campbell, C. T. *Nat. Chem.* **4**, 597-598 (2012)). We overlooked these parts and explained SMSI, and thanks to the reviewer's careful comments, we revised the description of the SMSI effect in Introduction section of the revised manuscript.

Referee comment #2. *While the manuscript presents a nice cartoon showing catalysis occurring at the metal-oxide interface, the experimental evidence for it was less convincing. Simply changing the pitch doesn't change the ratio of interface to metal sites – the width of the wires needs to be changed.*

Reply #2: We are so grateful that the reviewer gave us an important comment on the ratio of metal-oxide interface that we have overlooked. When Pt, Pd and Ni nanocatalysts were supported by CeO₂, there was a previous study in which the CO oxidation activity increased as the size of the nanoparticles decreased, because the interfacial sites (*i.e.*, perimeter, corner atoms) increased as the particle size decreased (Cargnello, M. *et al. Science* **341**, 771-773 (2013)). Thus, as pointed out by the reviewer, changing the width of the nanowire changes the density of interfacial sites, so we can control the ratio of interface to metal sites. As we have already mentioned in the manuscript, in fact the width of Pt nanowires deposited at 50 nm and 35 nm pitch was 20 nm and 15 nm, respectively. Thus, the ratio of interface to metal sites increased in deposited Pt nanowires with 15 nm rather than 20 nm in width, and the selectivity in chemical reactions increased due to the more formed metal-oxide interface. On Pt nanowire arrays with smaller width, due to the increased selectivity, creation of hot electrons were promoted, thereby the chemcurrent yield in catalytic nanodevices also increased.

Additionally, in order to further observe the selectivity and hot electron excitation changes as a function of the width control of Pt nanowires, we fabricated the Pt nanowire arrays by increasing the width to 50 nm and depositing it on TiO₂ support to produce a catalytic nanodevice. As shown in SEM image below (Fig. R3a), it was confirmed that Pt nanowires were deposited on TiO₂ support with a width of 50 nm and a pitch of 200 nm, and the *I-V* curve measurement showed that the Schottky nanodevice was well-fabricated with Schottky barrier height of 0.89 eV (Fig. R3b). As in the previous experiment with Pt nanowires with width 15 nm and 20 nm, the steady-state chemcurrent with increasing the reaction temperature for three cycles could be measured on the fabricated Pt nanowires (width 50 nm)/TiO₂ catalytic nanodiode under methanol 4 Torr and oxygen to 760 Torr, and the thermal stability of the nanodevice under reaction condition was also confirmed by checking the unchanged electrical properties through the measured *I-V* curve (Fig. R4a and

R4b). In addition, both catalytic activity and selectivity were measured to confirm the metal-oxide interface effect at increased width of Pt nanowires (Fig. R4c).

As the reviewer mentioned, the selectivity and hot electron generation according to the width of Pt nanowires were compared under methanol 4Torr and oxygen to 760 Torr, as shown in Fig. R5. Compared to Pt film, selectivity increased when Pt nanowires were deposited on TiO₂ support, and it was confirmed that selectivity was enhanced as width of the Pt nanowires decreased. As the width of Pt nanowires on TiO₂ became smaller, the ratio of interface to metal sites became larger, which enhanced selectivity to methyl formate formation owing to the increased nanoscale Pt-TiO₂ interfacial sites (Fig. R5d). Thus, when comparing the chemicurrent yield detected in catalytic nanodevices according to the width of Pt nanowires, it can be seen that the chemicurrent yield increased (*i.e.*, higher efficiency for hot electron generation) as the width of nanowires decreased, similar to the trend in selectivity (Fig. R5e). Accordingly, we can conclude that as the width of the Pt nanowires decreased, higher selectivity to methyl formate was observed due to the increased Pt-TiO₂ interface, thereby enhancing hot electron excitation in catalytic nanodiodes. These results show that the nanoscale metal-oxide interface plays an important role in determining selectivity and hot electron generation in methanol oxidation and that the concentration of this interface has a significant effect. The width dependence of both selectivity and hot electron excitation of these TiO₂ supported nanowires indicates that the methanol oxidation reaction primarily occurs at the metal-oxide interface. Thus, the well-controlled width of oxide supported nanowires through our lithography technique can easily tune the ratio of metal-oxide interface, which changed the chemical reaction significantly. We added all of these additional experimental results and descriptions (regarding the tuning of interface ratio by width control of nanowires) to the revised manuscript and supplementary information, and revised the figures and overall results (see Fig. 3, and Supplementary Figs. 12-16) of the manuscript by focusing on the width control rather than the pitch of Pt nanowires for changing the concentration of metal-oxide interface by reflecting the reviewer's comment.

Fig. R3 | a. SEM image of Pt nanowire arrays deposited on TiO₂ support with width of 50 nm.

b. *I-V* characteristics obtained from the Pt nanowires (width 50 nm)/TiO₂ catalytic nanodiode, which is obtained for a Schottky barrier height of 0.89 eV, and an ideality factor of 1.37, respectively.

Fig. R4 | **a.** Chemicurrent on a Pt nanowires (width 50 nm)/TiO₂ catalytic nanodiode measured under 4 Torr of methanol with O₂ to 760 Torr with elevated reaction temperature. **b.** *I-V* curves of the Pt nanowires (width 50 nm)/TiO₂ catalytic nanodiode before and after three cycles of catalytic reaction. **c.** TOF for production of CO₂ and methyl formate and selectivity to methyl formate on Pt nanowires (width 50 nm)/TiO₂ during 4 Torr of methanol and 760 Torr of O₂ with different reaction temperatures.

Fig. R5 | Selectivity and chemicurrent yield on nanowire arrays with different pitches and stacking compositions. SEM images of **a.** Pt nanowires/TiO₂ with a width of 15 nm, **b.** Pt nanowires/TiO₂ with a width of 20 nm and **c.** Pt nanowires/TiO₂ with a width of 15 nm, **b.** Pt nanowires/TiO₂ with a width of 50 nm. Scale bars are 200 nm. **d.** Selectivity to methyl formate under methanol oxidation on the Schottky nanodiodes at different temperatures with different widths. **e.** Chemicurrent yield for methanol oxidation on the catalytic nanodiodes of width 15 (Pt nanowires/TiO₂ with a width of 15 nm), width 20 (Pt nanowires/TiO₂ with a width of 20 nm), width 50 (Pt nanowires/TiO₂ with a width of 50 nm), and Pt film (Pt film/TiO₂) measured both at 333 K and 343 K.

Referee comment #3. *It is not clear what the Pt film/TiO₂ is or how it differs from the nanowire arrays. This is critical as this provides a possible control where there is little or no exposed interface. The film shows reasonable electrical activity, which suggests that the higher activity of the nanowires is only a Pt surface area effect and therefore that the reaction occurs on the Pt (and not at the interface).*

Reply #3: We thank the reviewer for this careful comment regarding the difference between the Pt film and Pt nanowires deposited on TiO₂. To show the morphology of the Pt film deposited on TiO₂, we added SEM images as shown below (Fig. R6). Since the Pt film was entirely flat and there were no cracks, it can be confirmed that the interface was not exposed because TiO₂ under the Pt film was not exposed to the gas environment. In our previous study obtained on Pt film/TiO₂ nanodevices, we confirmed by atomic force microscopy that the roughness of the Pt film is very small and overall uniform (Lee, S. W. *et al. ACS Catal.* **9**, 8424-8432 (2019)). As the reviewer pointed out, there was a reasonable chemicurrent density in Pt film/TiO₂ nanodiode, but it can be said that the reaction occurred completely on the Pt surface in the Pt film because the interfacial sites were not exposed to the methanol oxidation environment. Selectivity was low in Pt film without Pt-TiO₂ interface (Fig. 2d), which resulted in a smaller chemicurrent activity than in Pt nanowires/TiO₂ nanodiode (Fig. 2e).

In addition, since all the experimental results (e.g., TOF, chemicurrent density and chemicurrent yield) we obtained were calibrated to the surface area of Pt, the surface area effect pointed out by the reviewer can be excluded. The reaction rates were reported as TOF in units of product molecules produced per Pt site per second, and the number of Pt sites for nanowire arrays was calculated by geometrical assumptions. The Pt nanowire was considered to be rectangular and standing upright on a planar oxide support similar to previous study (Contreras, A. *et al. Cat. Lett.* **111**, 5-13 (2006)). Furthermore, if the higher activity in Pt nanowires/TiO₂ was due to the Pt surface area rather than the interface, the selectivity in Pt nanowires/SiO₂ should have been similar to that in Pt nanowires/TiO₂. However, when Pt nanowires were deposited on SiO₂, a non-reducible support with little metal-oxide interface effect (*i.e.*, support with weak Lewis acidity), the selectivity was lower than that of Pt nanowires/TiO₂, and the selectivity in Pt nanowires/SiO₂ was similar to that of Pt film/TiO₂ where no interfacial sites were exposed (Supplementary Fig. 17a). In response to all of the

comments, we believe that when Pt nanowires were deposited on TiO₂ support, it can be concluded that Pt-TiO₂ interfacial sites were formed to enhance selectivity, thereby increasing hot electron generation. As pointed out by the reviewer, we added all of these additional experimental results and explanations (regarding morphology of Pt film/TiO₂ and calibration with Pt surface area in Pt nanowires) to the revised manuscript.

Fig. R6 | SEM images of Pt nanowire arrays deposited on TiO₂ support using a self-assembled block copolymer with width of **a.** 15 nm, **b.** 20 nm, **c.** 50 nm, and **d.** Pt film.

Reply to Reviewer 2's report

Referee comment #0. *In this manuscript, the author construct Pt nanowires/TiO₂ catalytic nanodiode to explore the properties of metal-oxide interface. The relation between catalytic activity and selectivity was related to the hot electron generation. DFT was also carried out to confirm the critical role of Pt-TiO₂ interface. We highly appreciate the new method to reveal the relationship between catalytic performance and hot electron generation. However, there was some cracks still present in this manuscript. If the author successfully address these problems, we will be glad to accept this paper. Thus, I recommend a minor revision.*

Reply #0: We thank the reviewer for the acknowledgement of the importance of this study and favorable recommendation for publication after minor revisions. We particularly appreciate this high evaluation, “*We highly appreciate the new method to reveal the relationship between catalytic performance and hot electron generation... If the author successfully address these problems, we will be glad to accept this paper.*”. To address the reviewer’s comments, we added some explanations regarding the effect of valence for Pt and active sites where hot electrons are generated. Detailed responses to the comments are given below.

Referee comment #1. *What is the relationship between the hot electron generation and the valence of Pt? where the hot electron generated from? Oxygen vacancies or metal Pt surface? If the interfacial oxygen vacancy promotes the generation of hot electrons?*

Reply #1: Thank you for this valuable insight. To observe if the valence state of Pt nanowires changes during the catalytic reaction, in situ $I-V$ curves of the Pt nanowires/TiO₂ catalytic nanodiode were measured before and after the reaction. Since the $I-V$ curve represents the state at the junction between the Pt nanowires and TiO₂, if the oxidation state of the Pt nanowires changes during the reaction, the $I-V$ curve will change. However, the in situ $I-V$ curves during the reaction were relatively unchanged (Supplementary Fig. 5), and it was assumed that the state of the Pt nanowires was maintained as a metallic state during the reaction. Furthermore, we also confirmed that the state of the Pt nanowires was not changed after reaction as it remained a metallic state confirmed by XPS. Therefore, it can be said that hot electrons are generated on metallic Pt nanowire surfaces.

When the Pt nanowire was deposited on TiO₂ support and the Pt-TiO₂ interfacial site was exposed to the chemical reaction environment, the mechanism of methanol oxidation reaction changed, as was confirmed by our DFT calculation (i.e., different activation barrier by changed adsorption configuration on interfacial sites). Therefore, selectivity to methyl formate production at Pt-TiO₂ interfacial sites increased. We note that in our previous research, we found that hot electrons are generated much more in the reaction to methyl

formate than CO₂ (Lee, S. W. *et al. ACS Catal.* **9**, 8424-8432 (2019)), thus it can be concluded that enhanced hot electron excitation obtained on Pt nanowires/TiO₂ indeed originates from their improved selectivity on nanoscale Pt-TiO₂ interfacial sites. We added all of these explanations in the revised manuscript.

Referee comment #2. *The reference is not up to date; And in order to help the author to deeply understand the interfacial synergistic mechanism, the author should read and cite the following references: J. Am. Chem. Soc. 2018, 140, 11241. ACS Catal. 2017, 7, 7600. ACS Catal. 2019, 9, 2707.*

Reply #2: We thank the referee for pointing this out. As suggested by the reviewer, we added the latest references. In addition, we added the discussion about interfacial synergistic mechanism by charge transfer on metal nanoparticles supported on TiO₂ support.

“There have been previous studies to increase catalytic performance by increasing coverage degree and electron density of metal nanocatalysts at interfacial sites when metal nanoparticles were supported by TiO₂⁵⁷⁻⁵⁹.”

We also added the following recommended citations in the manuscript:

- 1) Xu, M. *et al.* Insights into interfacial synergistic catalysis over Ni@TiO_{2-x} catalyst toward water–gas shift reaction. *J. Am. Chem. Soc.* **140**, 11241–11251 (2018).
- 2) Xu, M. *et al.* TiO_{2-x}-modified Ni nanocatalyst with tunable metal–support interaction for water–gas shift reaction. *ACS Catal.* **7**, 7600–7609 (2017).
- 3) Liu, N. *et al.* Au^{δ-}-O_v-Ti³⁺ Interfacial Site: Catalytic Active Center toward Low Temperature Water Gas Shift Reaction. *ACS Catal.* **9**, 2707–2717 (2019).

Reviewers' comments:

Reviewer #1 (Remarks to the Author):

I have read the revised manuscript, but my main concerns are not addressed well. The authors did not re-organize the manuscript, they simply added more text to make it even more confusing.

By largest concern is that the authors have essentially assumed that the reactivity differences are due to "hot electrons" but still do not bother to prove this. Simply attributing any reactivity differences to "hot electrons" is not the same as proving that they are the cause. In looking at the new data, it is not even clear how important the interface is to the reactivity and selectivity. Again, a lack of control experiments make this difficult to determine.

Notably, in the new figure, the reactivity between the nanowires and the covered titania (panels a and b) is essentially the same. If the reactivity is the same, how can it be attributed to hot electrons? Similarly, the selectivity differences are not very large: an increase from 20% to 30%. This could simply be 25 +/- 5%, which is not a significant difference. Since these are %, the numbers can be deceiving. Plots of selectivity vs. conversion should have been prepared to more clearly show the differences and assess these differences.

Ultimately, the two plots do not look very different. Conversion vs. time is almost identical, so the two catalysts have the same activity. The selectivity differences appear to be due to lower production of CO₂. This could be simply due to CO₂ adsorption on the TiO₂. No control experiment adding TiO₂ to the Pt sheet was performed to check this.

The argument about TiO₂ being a Lewis acid is also probably not correct. While it can be a Lewis acid, unless care is taken to uncover those sites, they are probably not present – especially when methanol is being formed.

Based on all of this, I cannot support publication of this work in Nature Communications

Reviewer #2 (Remarks to the Author):

Recommendation: Publish in Nature Com without further revision!

Comment to the Author:

The author is very satisfied with all the answers. I am pleased to publish this paper without further revision!

Reply to Reviewer 1's report

Referee comment #1: *I have read the revised manuscript, but my main concerns are not addressed well. The authors did not re-organize the manuscript, they simply added more text to make it even more confusing.*

Reply #1: We thank the reviewer for their meticulous comments on organization to improve the quality of this manuscript. **Figs. 1, 2, and 4** were revised to further emphasize the reader's easy understanding of this study and the importance of the metal-oxide interface. First, in **Fig. 1**, the manufacturing process of Pt nanowires/TiO₂ nanodiode was moved to **Supplementary Fig. 1**, and an excitation process of hot electrons by exothermic catalytic reaction and a scheme for detecting this reaction-induced hot electron were included. In addition, a current signal of hot electrons generated by methanol oxidation reaction was moved to **Fig. 1**.

In **Fig. 2**, the results were reconstructed mainly to prove and emphasize the effect of Pt-TiO₂ interface (see **Fig. R1**). We added the plots of selectivity vs. conversion suggested by the reviewer to more clearly show the differences in selectivity between Pt nanowires/TiO₂ and Pt film/TiO₂ and emphasized the effect of the Pt-TiO₂ interface once again. In addition, in **Fig. 2**, to further prove and emphasize the effect of Pt-TiO₂ interface on selectivity, we added the results of our control experiment of depositing TiO₂ nanowires on the Pt sheet (i.e., inverse structure), which changed the stack order of nanowires. As mentioned in the previous version of the manuscript, since the width of the TiO₂ nanowires/Pt was the same as the previous Pt nanowire/TiO₂ structure, the ratio of Pt-TiO₂ interface was also the same for both structures, resulting in the same selectivity in both systems. Instead of the Pt-TiO₂ interface, we included the results of our control experiment of depositing the Pt nanowire arrays on a non-reducible SiO₂ support known to have little metal-support interface effect. In the Pt nanowires/SiO₂, the selectivity was not high, coming close to that of the Pt film/TiO₂. From this control experiment, it was found that the Pt-TiO₂ interface has a greater effect on selectivity enhancement than the Pt-SiO₂ interface. The results of these control experiments were in the previous Supplementary Figure, but they have been moved to **Fig. 2** to further emphasize the effect of the Pt-TiO₂ interface.

Furthermore, to theoretically emphasize that the Pt-TiO₂ interface plays an important role in increasing the selectivity of methyl formate production, the pathways in which CO₂ and methyl formate were generated in Pt nanorod/TiO₂(110) and investigated by theoretical simulation were added in **Fig. 4c**. CO₂ was generated on the Pt surface, and methyl formate was formed by reacting the intermediate present on TiO₂ with OH present on Pt; the importance of the Pt-TiO₂ interface in increasing the selectivity of methyl formate production could also be theoretically proven.

Therefore, in the revised manuscript, we mention that when the Pt-TiO₂ interface was formed in Pt nanowires/TiO₂, selectivity toward methyl formate increased due to the metal-support interaction, and this change in selectivity enhanced the reaction-induced hot electron excitation. Overall, we added data interpretation and control experiments to further support the part of the study that we would like to claim, and we have completely re-organized the manuscript. Detailed responses to the reviewer’s comments are given below.

Fig. R1 | Selectivity and hot electron generation on Pt nanowires/TiO₂ and Pt film/TiO₂ nanodiode. Comparison of **a** total reactivity and **b** partial oxidation selectivity under methanol oxidation for two Schottky nanodiodes of Pt film and Pt nanowires supported on TiO₂ at different temperatures. **c** Plot of selectivity according to methanol conversion on the catalytic nanodiodes of Pt film/TiO₂ and Pt nanowires/TiO₂ measured at 343 K. **d** Chemicurrent yield associated with the efficiency for hot electron flow obtained from the Pt film/TiO₂ and Pt nanowires/TiO₂ catalytic nanodiodes during methanol oxidation reaction. **e** Selectivity to methyl formate and **f** chemicurrent yield under methanol oxidation on the Schottky nanodiodes with different composition or stack order of nanowires. All results were obtained under methanol 4 Torr and O₂ to 760 Torr.

Referee comment #2. *By largest concern is that the authors have essentially assumed that the reactivity differences are due to “hot electrons” but still do not bother to prove this. Simply attributing any reactivity differences to “hot electrons” is not the same as proving that that they are the cause.*

Reply #2: It seems that the reviewer misunderstood our study. This work *never mentioned* that reactivity or selectivity changes ‘*due to*’ hot electrons. The reviewer also pointed this out in the last referee comment, and we responded in the last reply letter. We would like to make clear that we are not attributing the reactivity differences to “hot electrons”. The focus of the current investigation is that reaction-induced hot electron generation and catalytic selectivity are influenced by nanoscale metal-oxide interfaces, as also indicated by the title of our manuscript, “Controlling hot electron flux and catalytic selectivity with nanoscale metal-oxide interfaces”. In the study, we showed that formed nanoscale Pt-TiO₂ interface gives rise to the higher selectivity of methyl formate, and hot electrons were thereby excited much more on the Pt-TiO₂ interface, since hot electron generation was affected by selectivity under methanol oxidation.

In our study, we will again emphasize the process of excitation of hot electrons and why the increase of hot electron generation is caused by selectivity. Hot electrons with an energy of 1-3 eV were excited by energy conversion when exothermic chemical reactions occurred in the catalyst surface (i.e., reaction-induced hot electron flux), and we could detect this flow of hot electrons in real time by using a metal-semiconductor catalytic nanodiode (see **Fig. R2a**). That is, an exothermic catalytic reaction occurs *first*, and hot electrons are generated *by* this reaction process (i.e., chemical energy conversion). The direct detection of hot electrons generated by the catalytic reaction process using catalytic nanodiode was first studied by Somorjai *et al.* It is possible to collect hot electrons if the size of the metal catalyst is close to the mean free path (~ 10 nm), as these electrons are transported across the metal–semiconductor junction without collision. When the potential barrier formed at the metal–semiconductor junctions (i.e., the Schottky barrier) is lower than the energy of the electrons chemically excited (1-3 eV) by the catalytic reaction, the Schottky barrier allows excited hot electrons to irreversibly transport through the metal-semiconductor junction. Once hot electrons arrive at the semiconductor by transfer, they cannot go back to the metal catalyst, which leads to irreversible, one-way hot electron transfer from the metal to the semiconductor. Therefore, metal–semiconductor catalytic nanodiodes allow the quick capture of energetic hot electrons before thermalization; the detected current is known as “*chemicurrent*” since the current of hot electrons is originated from the chemical reaction on the metal catalyst surface. In this study, the chemically excited hot electrons generated from methanol oxidation were enough to irreversibly overcome the Schottky barrier at the Pt nanowire arrays/TiO₂ junction when the excess chemical energy was higher than the Schottky barrier height of the nanodiode (i.e., obtained sufficient energy) and could be detected as a steady-state current of hot electron flow (see **Fig. R2a**). The fact that this chemicurrent was actually produced by a catalytic reaction is the difference between the

detected current when the reaction occurs and when it does not. As shown in **Fig. R2b**, a definite deviation between the currents measured with (*i.e.*, methanol oxidation condition) and without (*i.e.*, pure oxygen condition) catalytic reaction was observed, and the difference in magnitude of the currents was clearly associated with hot electron generation by the catalytic methanol oxidation on the catalyst. The hot electron excitation process by chemical energy conversion in the exothermic catalytic reaction and the technique of detecting it by using metal-semiconductor Schottky nanodiode were already described in the first paragraph of the introduction part in our manuscript.

Planar metal on a semiconductor nanodiode was used in previous studies. But here, we fabricated nanodevices by depositing Pt nanowires on TiO_2 to reveal the effect of the Pt- TiO_2 interface. The reaction we used was methanol oxidation, which produced CO_2 and methyl formate. In our previous model study, we directly detected hot electrons from a methanol oxidation reaction on a Pt film/ TiO_2 nanodiode. In this previous study, we found that the selectivity of methyl formate production in the methanol oxidation reaction is correlated with hot electron excitation due to higher exothermicity. In experimental results, hot electrons were rarely detected in conditions where only CO_2 was generated in methanol oxidation, and it was found that reaction-induced hot electron excitation was boosted when methyl formate was produced. This could also be verified by the fact that more exothermic energy was generated in methyl formate production through DFT calculation (see Lee, S. W. *et al. ACS Catal.* **9**, 8424-8432 (2019)). That is, the reaction-induced hot electrons here are generated by non-adiabatic electronic excitation as the exothermic energy in the chemical reaction is converted, and thus, hot electron generation is increased in the reaction to the methyl formate, a pathway with high exothermicity. Therefore, as selectivity increases, hot electron excitation increases. We have added a description of this hot electron excitation process to the revised manuscript. In addition, to prevent confusion, we emphasized several times in the revised manuscript that this hot electron was induced by a chemical reaction, and that this reaction-induced hot electron excitation increased due to the increase of selectivity. The key point of this study is that the chemicurrent yield increased *due to* the increase in selectivity by the Pt- TiO_2 interface in the Pt nanowires/ TiO_2 catalyst.

Fig. R2 | **a** Energy band diagram for Schottky nanodiode of the Pt nanowires supported on TiO₂. Hot electrons excited from the surface chemical reaction can be detected as a steady-state chemicurrent by charge transfer if their excess chemical energy is large enough to overcome the Schottky barrier of the Pt-TiO₂ junction. **b** Current density associated with the methanol oxidation measured on the Pt nanowires/TiO₂ with increased reaction temperature. The differences in the magnitude of the currents measured with and without catalytic reaction were associated with reaction-induced hot electrons generated on the catalytic nanodiode (*i.e.*, net chemicurrent).

Currently, in this paper, the catalytic nanodiode was prepared using Pt nanowires instead of planar Pt film, and it is shown that the selectivity of methyl formate production increased due to the nanoscale Pt-TiO₂ interface formed between Pt nanowires and TiO₂. Similar to the previous planar Pt film/TiO₂ results, in our Pt nanowires/TiO₂ nanodiode system, the reaction condition in which only CO₂ was produced (*i.e.*, methanol 1, 2 Torr and O₂ to 760 Torr) showed a very low chemicurrent yield (*i.e.*, a low efficiency of hot electron generation on the catalyst surface), and that a lot of hot electrons were excited when methyl formate was produced (*i.e.*, methanol 4 Torr and O₂ to 760 Torr) (see **Figs. R3 and R4** below). Thus, it was observed that the chemicurrent yield value significantly increased, due to the increased selectivity toward methyl formate formation. Here, the chemicurrent yield is the number of hot electrons generated when one reaction occurs, that is, this term indicates the *efficiency of reaction-induced hot electron generation*, since it is a value obtained by dividing chemicurrent by reactivity. Thus, the increase in chemicurrent yield in Pt nanowires/TiO₂ compared to Pt film/TiO₂ (under methanol 4 Torr and O₂ to 760 Torr) means that hot electron excitation occurred more efficiently on Pt nanowires/TiO₂, because the selectivity of methyl formate production increased. The reviewer understands that selectivity increased owing to hot electrons, but the order is *reversed* in this study. The selectivity to methyl formate increased due to the formed nanoscale Pt-TiO₂ interface, and this enhancement of selectivity increased hot electron excitation (*i.e.*, chemicurrent yield).

As shown in **Fig. R4**, under methanol 1 and 2 Torr conditions, no significant difference in the chemicurrent yield was found for Pt nanowires/TiO₂ and the Pt film/TiO₂ nanodiode. As mentioned earlier, because of excess oxygen conditions in the 1 and 2 Torr environments of methanol, methyl formate was not produced, and only the Pt-TiO₂ interface produced CO₂ like the Pt film; thus, the chemicurrent yield was determined only by the production rate of CO₂. However, in the methanol 4 Torr environment, Pt nanowires deposited on TiO₂ showed higher chemicurrent yield than Pt film. In this environment, a partial oxidation reaction took place, methyl formate formed, and the chemicurrent yield *was determined by* partial oxidation selectivity. Thus, the Pt nanowires showed higher selectivity due to the Pt-TiO₂ interface, *resulting in* higher chemicurrent yield. Thus, it can be seen that chemicurrent yield, which represents the efficiency of hot electron excitation, was greatly influenced by selectivity to methyl formate, *not reactivity*. Therefore, we used the results

obtained in methanol 4 Torr and O₂ to 760 Torr, which was a condition in which methyl formate was produced when comparing results in different nanodiodes.

Again, we note that in our previous research, we found that hot electrons are generated much more in the reaction to methyl formate than CO₂, and it can thereby be concluded that the enhanced hot electron excitation obtained on Pt nanowires/TiO₂ does indeed *originate from* their increased selectivity on nanoscale Pt-TiO₂ interfacial sites. We added this emphasis to the revised manuscript so that other readers would not be confused. In addition, the previous manuscript's introduction mentioned previous studies on charge transfer's affect on reaction rate; as hot electrons are generated *by* catalytic reaction in our study, and this distinction could be confusing to other readers as it was to the reviewer, we

have removed these references from the revised manuscript.

Fig. R3 | Measurements of net chemicurrent excited by catalytic methanol oxidation on Pt nanowires/TiO₂ Schottky nanodiode varying with the partial pressure of methanol (1-4 Torr) in 760 Torr of O₂ at different temperatures, showing the methanol partial pressure dependence on hot electron generation.

Fig. R4 | Plot of chemicurrent yield as a function of partial pressures of methanol detected on both the Pt film/TiO₂ and Pt nanowires/TiO₂ catalytic nanodiodes at **a** 333 K and **b** 343 K.

Referee comment #3. *In looking at the new data, it is not even clear how important the interface is to the reactivity and selectivity. Again, a lack of control experiments make this difficult to determine.*

Reply #3: Regarding this point, we have demonstrated that the nanoscale interface plays an important role in selectivity through many control experiments, including (i) comparison between Pt nanowires/TiO₂ and Pt film/TiO₂, (ii) control of interface ratio through width control of nanowires and effect of this width on selectivity, (iii) comparison between Pt nanowires/TiO₂ and Pt nanowires/SiO₂, and (iv) comparison between Pt nanowires/TiO₂ and TiO₂ nanowires on Pt thin film. Based on these experiments, we can conclude that nanoscale Pt-TiO₂ interface influences the catalytic selectivity as well as the hot electron generation.

First, as previously pointed out by the reviewer, changing the width of the nanowire changes the density of interfacial sites, which we can use to control the ratio of interface to metal sites. In other words, a greater ratio of Pt-TiO₂ interface to metal sites could be modeled by decreasing the width of Pt nanowires. The detailed experimental results of each Pt nanowires/TiO₂ nanodiode with different widths are shown in **Supplementary Fig. 18-23**. Here, we found that, as the width of the Pt nanowires decreased from 50 nm to 15 nm, higher selectivity to methyl formate was observed due to the increased Pt-TiO₂ interface (i.e., ratio of interface to metal sites) (see Fig. R5a). The selectivity increased due to the increased nanoscale Pt-TiO₂ interface at the reduced width, and the reaction-induced hot electron excitation increased due to the enhanced selectivity. The width dependence of selectivity of the oxide supported nanowires indicates that the catalytic methanol oxidation reaction primarily occurs at the Pt-TiO₂ interfacial sites. Here again, it is emphasized that when the width decreases, the selectivity does not increase due to hot electrons; conversely, the

selectivity enhances and causes an increase in hot electron generation (see **Fig. R5b**). Therefore, since the structure exhibiting smaller width has a greater ratio of nanoscale Pt-TiO₂ interface to metal sites, it exhibits higher selectivity, thereby increasing reaction-induced hot electron excitation in methanol oxidation.

Fig. R5 | **a** Selectivity to methyl formate under methanol oxidation on the Schottky nanodiodes at different temperatures with different widths. **b** Chemicurrent yield for methanol oxidation on the catalytic nanodiodes of width 15 (Pt nanowires/TiO₂ with a width of 15 nm), width 20 (Pt nanowires/TiO₂ with a width of 20 nm), width 50 (Pt nanowires/TiO₂ with a width of 50 nm), and Pt film (Pt film/TiO₂) measured both at 333 K and 343 K. All results were obtained under methanol 4 Torr and O₂ to 760 Torr.

In particular, in order to prove the effect of the Pt-TiO₂ interface once again, instead of the Pt-TiO₂ interface, we conducted further control experiments that changed the oxide support of nanowires by forming a Pt-SiO₂ interface, which is known to have little metal-support interaction (*i.e.*, depositing Pt nanowires on SiO₂ support). When Pt nanowires were deposited on SiO₂ support, selectivity was not high, and the results were similar to those of Pt thin films without metal-oxide interface exposure (see **Fig. R6a**). Hence, the size and shape effect of the Pt nanowires could be excluded from the results of similar selectivity for Pt nanowires/SiO₂ and Pt film. In other words, the formed nanoscale Pt-TiO₂ interfacial sites in Pt nanowires/TiO₂ have a much greater effect on the methanol oxidation reaction than the effect of the structure of the Pt nanowire. Moreover, we can conclude that the Pt-TiO₂ interface has a greater effect on selectivity enhancement than the Pt-SiO₂ interface. Hence, the enhancement of selectivity only when Pt nanowires were deposited on TiO₂, rather than the non-reducible SiO₂ support, may well be strong evidence of the claim that methanol oxidation reactions occur mainly at nanoscale Pt-TiO₂ interfacial sites. Through this control

experiment, in which Pt nanowires were deposited on SiO₂, a non-reducible support, the importance of the Pt-TiO₂ interface in our Pt nanowires/TiO₂ system was once again confirmed.

We conducted further control experiment that changed the stacking order of nanowires to further investigate the effect of the Pt-TiO₂ interface (see **Supplementary Fig. 15** for details). In this scenario, TiO₂ nanowires were deposited on a Pt thin film to form an inverse structure (*i.e.*, TiO₂ nanowires/Pt) and the same nanoscale Pt-TiO₂ interface was formed. Here, the width of the TiO₂ nanowires/Pt was the same as the width of the existing Pt nanowires/TiO₂, thus, the ratio of Pt-TiO₂ interfacial sites were the same in both two-catalyst systems. When comparing the results in these two systems, the selectivity of methyl formate was nearly the same, proving that the same amount of interface ratio could not change the selectivity (see Fig. R6a); if the same ratio of metal-oxide interfacial sites formed, it could be seen that the stack order had little effect on determining selectivity. Since the selectivity was similar, the efficiency of reaction-induced hot electron generation (*i.e.*, chemicurrent yield) was similar for two catalytic nanodiodes (see **Fig. R6b**). Therefore, it was confirmed that the Pt-TiO₂ interface plays an important role, regardless of the stacking order.

Fig. R6 | **a** Selectivity to methyl formate and **b** chemicurrent yield under methanol oxidation on the Schottky nanodiodes at different temperatures with different oxide support or stack order of nanowires.

We have demonstrated through these control experiments that the nanoscale Pt-TiO₂ interface plays an important role in selectivity in the methanol oxidation reaction. (1. Comparison of Pt nanowires/TiO₂ and Pt film/TiO₂, 2. Control of metal-oxide interface ratio through width control of Pt nanowires and effect of this width on selectivity, 3. Comparison of Pt nanowires/TiO₂ and Pt nanowires/SiO₂, 4. Compared to the system in which the

stacking order of nanowires was reversed). We would like the reviewer to see the results of these control experiments once again. In addition, in order to further emphasize the comparison of Pt nanowires/TiO₂ with Pt nanowires/SiO₂ and TiO₂ nanowires/Pt film systems in this paper (*i.e.*, to show the importance of the Pt-TiO₂ interface more clearly), the results of the previous Supplementary Figure were added to the main **Fig. 2** in the revised manuscript.

Referee comment #4. *Notably, in the new figure, the reactivity between the nanowires and the covered titania (panels a and b) is essentially the same. If the reactivity is the same, how can it be attributed to hot electrons?*

Reply #4: As the reviewer said, the total TOF of Pt film/TiO₂ and Pt nanowires/TiO₂ (*i.e.*, the sum of the TOFs of CO₂ and methyl formate production) is almost the same, as shown in **Fig. R7**. It has already been pointed out in the manuscript that we have similar total TOFs in both systems (*i.e.*, Pt-TiO₂ interface does not affect *total reactivity*). We would like to make clear that we are not attributing the reactivity differences to “hot electrons”. The focus of the current investigation is that the reaction-induced hot electron generation as well as the catalytic selectivity is influenced by the nanoscale metal-oxide interfaces, as also indicated by the title of our manuscript “Controlling hot electron flux and catalytic selectivity with nanoscale metal-oxide interfaces”.

Since the reaction we used was methanol oxidation, which is a multi-path reaction in which two products are produced, rather than a one-path reaction, it is correct to compare in terms of *selectivity* rather than *reactivity*. In this methanol oxidation reaction, it is important to reduce the production of CO₂, the main cause of global warming, and increase the production of methyl formate, a valuable chemical. Therefore, what we argue in this paper is that the production of CO₂ was reduced and the production of methyl formate was enhanced by the formed nanoscale Pt-TiO₂ interface in Pt nanowires/TiO₂. In fact, the production of CO₂ was reduced and the production of methyl formate was increased by the Pt-TiO₂ interface in Pt nanowires/TiO₂, so the total TOF was similar to that of Pt film/TiO₂ (*i.e.*, Coincidentally, methyl formate production increased as CO₂ decreased.). Owing to the formed Pt-TiO₂ interface, reduced CO₂ production and increased methyl formate production can also be verified through our DFT calculation (*i.e.*, when the Pt-TiO₂ interface is formed compared to Pt(111), the activation barrier of the path to CO₂ increases and the barrier of the path to methyl formate decreases (see **Fig. R8**)). Hence, it can be said that Pt nanowires/TiO₂ was a more efficient catalyst in that the selectivity of methyl formate production in Pt nanowires/TiO₂ was significantly increased compared to Pt film/TiO₂.

Fig. R7 | Comparison of catalytic activity under methanol oxidation for two Schottky nanodiodes of Pt film and Pt nanowires supported on TiO₂ at different temperatures.

Fig. R8 | Calculated activation barrier for Pt nanorod on $\text{TiO}_2(110)$ and Pt(111). Optimized free-energy profiles for **a** methyl formate and **b** CO_2 formation path with reaction barriers (ΔG^\ddagger). Barriers with red and blue indicate the RDS of each path for Pt nanorod structure on $\text{TiO}_2(110)$ and Pt(111), respectively. **c** Schematic drawing showing formation of CO_2 and methyl formate in methanol oxidation on Pt nanorod/ $\text{TiO}_2(110)$. Species marked with an asterisk (*) are adsorbed on the surface. The grey, blue, brown, red, and yellow balls indicate Pt, Ti, C, O, and H, respectively.

We also compared Pt nanowires/TiO₂ and Pt film/TiO₂ in a hydrogen oxidation reaction, a one-path reaction in which only one product appears, as a comparative control experiment (see **Fig. R9**). The reaction conditions were H₂ 4 Torr and O₂ to 760 Torr, similar to the environment in methanol oxidation, and comparing the results in the two systems yielded nearly the same TOF (*i.e.*, similar reactivity). This confirmed that there was little metal-oxide interface effect on the reactivity in the hydrogen oxidation reaction where selectivity does not appear, establishing that the Pt-TiO₂ interface had a greater effect on the selectivity in the multi-path reaction than the reactivity.

Fig. R9 | **a** TOF for production of H₂O under 4 Torr of hydrogen and 760 Torr of O₂ on the Pt film/TiO₂ and Pt nanowires/TiO₂ with different reaction temperatures. **b** Arrhenius plots obtained from measurement of TOF on the Pt film/TiO₂ and Pt nanowires/TiO₂ under 4 Torr of hydrogen and 760 Torr of O₂.

Fig. R10 | Current measured on **a** Pt nanofilm/TiO₂ and **b** Pt nanowires/TiO₂ catalytic nanodiode in 4 Torr of hydrogen and 760 Torr of O₂ with elevated reaction temperature. **c** Chemicurrent yield on Pt film/TiO₂ and Pt nanowires/TiO₂ catalytic nanodiode with different reaction temperatures under hydrogen oxidation.

As emphasized above, reaction-induced hot electron excitation (*i.e.*, chemicurrent) is *caused by exothermic energy* in catalytic reactions, and catalytic reactions are *not altered by* hot electrons. We found in previous research that the selectivity of methyl formate production in the methanol oxidation reaction further boosts hot electron excitation. That is, chemicurrent is more affected by selectivity than reactivity. Hence, in our system, when the Pt-TiO₂ interface was formed in Pt nanowires/TiO₂, selectivity was enhanced by the metal-support interaction (see **Fig. R11a** below), and this enhancement of selectivity could increase the chemicurrent yield (see **Fig. R11b** below). Even if the total TOF (*i.e.*, reactivity) in Pt nanowires/TiO₂ and Pt film/TiO₂ are similar, it has no effect on chemicurrent. As mentioned above, in the hydrogen oxidation reaction in which selectivity does not actually appear (*i.e.*, only reactivity), the change in reactivity and chemicurrent yield by the formed Pt-TiO₂ interface did not appear (see **Figs. R9 and R10**). Therefore, it can be concluded that selectivity has a much greater effect than reactivity in determining the efficiency of reaction-induced hot electron excitation. Thus, the enhancement of hot electron excitation was attributed to the increased selectivity forming methyl formate. We added to the revised manuscript that Pt nanowires/TiO₂ was a more efficient catalyst for selective reaction producing methyl formate when compared to Pt film/TiO₂, because, although the total TOF is similar, the selectivity was greatly enhanced. We have never mentioned that this selectivity change was attributed by hot electrons, as, again, our experimental system was the *opposite concept*. Thus, our study is the detection of hot electrons from chemical energy conversion by exothermic methanol oxidation reaction with using a metal-semiconductor catalytic nanodiode, and the excitation of this hot electron is attributed by the selectivity of methanol oxidation. In the revised manuscript, we frequently emphasized this concept by mentioning the reaction-induced hot electron excitation.

Fig. R11 | **a** Comparison of partial oxidation selectivity under methanol oxidation for two Schottky nanodiodes of Pt film and Pt nanowires supported on TiO₂ at different temperatures. **b** Chemicurrent yield associated with the efficiency for hot electron flow obtained from the Pt film/TiO₂ and Pt nanowires/TiO₂ catalytic nanodiodes during methanol oxidation reaction.

Referee comment #5. *Similarly, the selectivity differences are not very large: an increase from 20% to 30%. This could simply be 25 +/- 5%, which is not a significant difference. Since these are %, the numbers can be deceiving.*

Reply #5: We strongly disagree with the reviewer on this point. The selectivity of methyl formate on Pt thin film/TiO₂ is 22.6 ± 0.6 %, while that on Pt nanowires/TiO₂ is 38.6 ± 0.6 % at 50 °C. Error scale in our measurement is less than 1%. Enhancement of selectivity is 70 %, which is significant.

In addition, the result of the control experiment (as shown in Fig. 3d or R5) shows the systematic trend that selectivity increases as the area of nanoscale metal-oxide interfaces increases. When the width of the Pt nanowire decreased to 15 nm (i.e., the larger the ratio of interface to metal sites), the observed enhancement of selectivity compared to Pt film/TiO₂ was even greater.

In order to show this point, we added the table of measured catalytic activity and calculated selectivity on tested catalysts to the revised supplementary information (see **Table R1 and R2** below).

Pt film/TiO ₂	1 st cycle	2 nd cycle	Average
Production of CO₂ (TOF)			
50 °C	0.036	0.047	0.042 ± 0.006
60 °C	0.137	0.119	0.128 ± 0.009
70 °C	0.365	0.341	0.353 ± 0.012
80 °C	0.997	0.773	0.885 ± 0.112
Production of methyl formate (TOF)			
50 °C	0.011	0.013	0.012 ± 0.001
60 °C	0.038	0.029	0.033 ± 0.004
70 °C	0.094	0.076	0.085 ± 0.009
80 °C	0.249	0.179	0.214 ± 0.035
Total TOF (molecules/site/s)			
50 °C	0.047	0.061	0.054 ± 0.007
60 °C	0.175	0.148	0.161 ± 0.014
70 °C	0.459	0.417	0.438 ± 0.021
80 °C	1.247	0.952	1.099 ± 0.148
Selectivity (%)			
50 °C	23.2	22.1	22.650 ± 0.550
60 °C	21.5	19.6	20.550 ± 0.950
70 °C	20.5	18.3	19.400 ± 1.100
80 °C	20.0	18.8	19.400 ± 0.600

Table R1 | Measured catalytic activity and calculated selectivity on Pt film/TiO₂.

Pt nanowires/TiO ₂	1 st cycle	2 nd cycle	Average
Production of CO₂ (TOF)			
50 °C	0.046	0.032	0.039 ± 0.007
60 °C	0.096	0.093	0.094 ± 0.002
70 °C	0.294	0.432	0.363 ± 0.069
80 °C	0.938	1.060	0.999 ± 0.061
Production of methyl formate (TOF)			
50 °C	0.030	0.020	0.025 ± 0.005
60 °C	0.042	0.046	0.044 ± 0.002
70 °C	0.113	0.182	0.148 ± 0.035
80 °C	0.363	0.397	0.380 ± 0.017
Total TOF (molecules/site/s)			
50 °C	0.076	0.052	0.064 ± 0.012
60 °C	0.138	0.139	0.138 ± 0.000
70 °C	0.407	0.614	0.511 ± 0.104
80 °C	1.301	1.457	1.379 ± 0.078
Selectivity (%)			
50 °C	39.2	38.0	38.600 ± 0.600
60 °C	30.3	33.2	31.750 ± 1.450
70 °C	27.7	29.7	28.700 ± 1.000
80 °C	27.9	27.3	27.600 ± 0.300

Table R2 | Measured catalytic activity and calculated selectivity on Pt nanowires/TiO₂.

Referee comment #6. *Plots of selectivity vs. conversion should have been prepared to more clearly show the differences and assess these differences.*

Reply #6: We would like to thank the reviewer for pointing this out. As mentioned earlier, since our system was a 2D batch reactor, all reaction data was obtained from a low conversion region (*i.e.*, under a kinetically controlled regime). As the reviewer suggested, to more clearly show the selectivity differences and assess these differences on Pt nanowires/TiO₂ and Pt film/TiO₂, the selectivity according to methanol conversion was plotted, as shown in **Fig. R12**. The selectivity of each catalyst according to methanol conversion was almost constant, which indicates that the reaction kinetics of methanol oxidation were constant within this comparable methanol conversion region. Even when comparing selectivity according to conversion, it can be seen that the selectivity of Pt nanowires/TiO₂ was increased compared to Pt film/TiO₂. In addition, the trend of selectivity according to the width of Pt nanowire was also clearly shown in all methanol conversion regions (see **Fig. R13**). That is, as the ratio of interface to metal sites increases (*i.e.*, the width decreases), selectivity increases due to the influence of the Pt-TiO₂ interface. From these results, we claim that selectivity increased in Pt nanowires compared to Pt film within the comparable methanol conversion region, and the trend according to nanowire width also proves that nanoscale Pt-TiO₂ interface plays an important role in determining selectivity. As suggested by the reviewer, we added these plots of selectivity vs conversion to the revised manuscript (**Fig. 2c, Supplementary Fig. 9 and 23**).

Fig. R12 | Plots of selectivity vs. methanol conversion on the catalytic nanodiodes of Pt film/TiO₂ and Pt nanowires/TiO₂ measured at **a** 343 K and **b** 353 K.

Fig. R13 | Plots of selectivity according to methanol conversion on the catalytic nanodiodes of width 15 (Pt nanowires/TiO₂ with a width of 15 nm), width 20 (Pt nanowires/TiO₂ with a width of 20 nm), width 50 (Pt nanowires/TiO₂ with a width of 50 nm), and Pt film (Pt film/TiO₂) measured at 343 K.

Referee comment #7. *Ultimately, the two plots do not look very different. Conversion vs. time is almost identical, so the two catalysts have the same activity.*

Reply #7: As we mentioned earlier, the methanol oxidation reaction has two products, CO₂ and methyl formate, and the total TOF (*i.e.*, the sum of TOF for CO₂ and methyl formate) in Pt film/TiO₂ and Pt nanowires/TiO₂ was similar. Thus, methanol conversion, which is the rate at which reactant methanol is converted to products, was similar in both systems. As explained earlier, this is because CO₂ production decreased and methyl formate production increased when the Pt-TiO₂ interface formed in the Pt nanowires/TiO₂ catalyst. Thereby, Pt nanowires/TiO₂ has higher selectivity to form methyl formate than Pt film/TiO₂. As mentioned above, it is correct to compare the selectivity of the desired product rather than comparing total reactivity or methanol conversion, since two products come out from the methanol oxidation reaction (*i.e.*, it is desirable to reduce the production of CO₂ and increase the production of methyl formate). In addition, through comparison of selectivity according to conversion in Pt nanowires and Pt film proposed by the reviewer, it can be seen that the overall selectivity of Pt nanowires/TiO₂ was higher than that of Pt film/TiO₂, which means that the catalytic properties of the two catalysts were different (see **Fig. R12**). Therefore, Pt nanowires/TiO₂ was a more efficient catalyst to increase selectivity forming methyl formate than Pt film/TiO₂.

Referee comment #8. *The selectivity differences appear to be due to lower production of CO₂. This could be simply due to CO₂ adsorption on the TiO₂. No control experiment adding TiO₂ to the Pt sheet was performed to check this.*

Reply #8: We thank the reviewer for this meticulous comment regarding the reason for the lower production of CO₂ in Pt nanowires/TiO₂ catalyst. As the reviewer mentioned, the lower production rate of CO₂ may be due to the adsorption of CO₂ on TiO₂, so we investigated the binding energy of CO₂ on the TiO₂ surface by DFT calculation on the Pt nanorod/TiO₂(110) model. As shown in **Fig. R14**, the calculated binding energy of CO₂ on the TiO₂ surface was 0.21 eV, which is an energy level that can be sufficiently desorbed even at room temperature (Nørskov, J. K. *et al.*, *Fundamental concepts in heterogeneous catalysis*. John Wiley & Sons, (2014)), thus the low production of CO₂ cannot be the adsorption of CO₂ on the surface of TiO₂. Moreover, in the DFT calculation on the Pt nanorod/TiO₂(110), HCO, which was the intermediate in the pathway of CO₂ formation, prefers adsorption to the Pt surface rather than TiO₂, and CO₂ formation occurs at the Pt site; thus, CO₂ adsorption in TiO₂ can also be excluded. Furthermore, since the reaction mechanism in which CO₂ was produced by methanol oxidation was revealed by theoretical calculation as an *Eley-Rideal* mechanism (CO_(g) + *O → CO_{2(g)}) in the Pt nanorod/TiO₂(110) model, there was no state in which CO₂ was adsorbed during the methanol oxidation reaction. Hence, it can be said that the decrease in production of CO₂ and the increase in production of methyl formate in Pt nanowires/TiO₂ was not due to the adsorption of CO₂ in TiO₂, but due to the *difference in activation barrier* investigated by our DFT calculation in Pt nanorod/TiO₂(110) and Pt(111) (see **Fig. R8**). These simulation results and explanations were added to the revised manuscript.

In addition, the influence of CO₂ adsorbed on two-dimensional TiO₂ surface in the gas measurement with gas chromatography in the steady-state condition is minor, considering the partial pressure of reactant gases of batch reactor is in the range of Torr. Regarding the point about the control experiment adding TiO₂ to the Pt sheet, we have already carried out the methanol oxidation reaction in the inverse structure, the control experiment mentioned by the reviewer, in which TiO₂ nanowires are deposited on the Pt sheet; the results of this control experiment are already in the manuscript (see **Fig. R6** and **Supplementary Fig. 15** for details). As previously explained, the inverse TiO₂ nanowires/Pt film structure was made by depositing TiO₂ nanowires on the Pt thin film with the same width of the nanowires as in the Pt nanowires/TiO₂ structure to measure catalytic reaction and chemi-current. Since the width of the two structures was the same, the proportion of Pt-TiO₂ interfacial sites formed was the same, and this results in almost the same selectivity in both systems. Owing to the almost identical selectivity in TiO₂ nanowires/Pt and Pt nanowires/TiO₂, chemi-current yields were also similar for both catalyst systems. Therefore, this control experiment in an inverse structure fabricated by placing TiO₂ nanowires on a Pt sheet can prove that the interface formed between Pt and TiO₂ greatly affects selectivity in methanol oxidation.

Fig. R14 | Calculation of binding energy when CO₂ is adsorbed on TiO₂(110) in Pt nanorod/TiO₂(110) model by DFT.

Referee comment #9. *The argument about TiO₂ being a Lewis acid is also probably not correct. While it can be a Lewis acid, unless care is taken to uncover those sites, they are probably not present – especially when methanol is being formed.*

Reply #9: We would like to thank the reviewer for pointing this out. We explained that the difference in selectivity between Pt nanowires/TiO₂ and Pt nanowires/SiO₂ (see **Fig. R6a**) was due to the difference in Lewis acidity of oxide support, reflecting the reviewer's previous review comment below.

“The different acid/base properties of silica and titania could also be driving this difference.”

However, as the reviewer said in this comment, we had not discussed in the previous version of the manuscript whether the TiO₂ support is a Lewis acid in methanol oxidation reaction conditions. Therefore, using DFT calculation, we investigated the charge transfer from the adsorbate molecule to the surfaces of oxide support when the reactant methanol molecule was adsorbed on TiO₂ and SiO₂ support to reveal the acid/base properties of these two supports during methanol oxidation. In the chemistry of Lewis acid/base properties on oxide surfaces, it has been defined as a Lewis acid when electron charge is received and a Lewis base when electron charge is lost (Metiu, H. *et al. J. Phys. Chem. C* **116**, 10439-10450 (2012)). The charge transfer by the interaction between adsorbate molecule-oxide was investigated using the Bader charge analysis simulation method, which is commonly used to

examine Lewis acidity of oxide support (Bader, R. *et al. Atoms in Molecules: A Quantum Theory*. Clarendon Press: Oxford England (1994) and Hirunsit, P. *et al. Chemphyschem* **19**, 2848-2857 (2018)).

As in the model designed for DFT calculation on the Au-TiO₂ interface in the Yates group (Green, I. X. *et al. Science* **333**, 736-739 (2011)), three atomic layers of Pt nanorod bonded on top of the rutile TiO₂(110), and the SiO₂ surface was used to model the Pt nanowires/TiO₂ and Pt nanowires/SiO₂. We used the interfacial structure of Pt-oxide support as shown in **Supplementary Fig. 16** to calculate the charge transfer between methanol adsorbate and oxide surfaces. And when the methanol was adsorbed on two oxide surfaces, the electron density on the methanol basis was calculated (*i.e.*, negative sign indicates that losing their electrons), yielding -0.07 when adsorbed on the TiO₂ surface and +0.01 when adsorbed on the SiO₂ surface (see **Fig. R15**). This result implies that when the adsorption of methanol occurs on the two oxide surfaces, electron transfer from methanol to surface occurs on the TiO₂ surface (*i.e.*, Lewis acid), and electron transfer from surface to methanol slightly occurs on the SiO₂ surface (*i.e.*, Lewis base). Therefore, from this Bader charge analysis, it was found that Lewis acidity of TiO₂ is stronger than that of the SiO₂ surface when methanol oxidation reaction occurs. Therefore, when Pt nanowires were deposited on the TiO₂ support, which has higher Lewis acidity than SiO₂, it showed higher partial oxidation selectivity. A previous study also indicated that the selectivity of methyl formate production in the methanol oxidation reaction increased as the Lewis acidity of the oxide support increased when the metal catalyst was introduced to the oxide support (Yang, Z.-Y. *et al. Catal. Today* (in press, 2019)). As a result, it has been confirmed that the Pt-SiO₂ interface had little metal-support interface effect compared to the Pt-TiO₂ interface, due to the different acid/base properties of SiO₂ and TiO₂. Hence, the increase in selectivity only occurred when Pt nanowires were deposited on TiO₂, which was a reducible support (*i.e.*, strong Lewis acidity), rather than non-reducible SiO₂, provide strong evidence of the claim that catalytic reactions occur mainly at Pt-TiO₂ interface sites. We are grateful to the reviewer for pointing this out, and we added these simulation calculation results and description to the revised manuscript.

Fig. R15 | Charge transfer from methanol adsorbate to oxide surfaces calculated by using Bader charge analysis simulation: **a** Pt nanorod/TiO₂, and **b** Pt nanorod/SiO₂.

REVIEWER COMMENTS

Reviewer #3 (Remarks to the Author):

The authors have added additional experimental and theoretical results to signify the role of metal-oxide interface. Nevertheless, as the previous reviewer mentioned there is significant room to improve the organization of the manuscript: for example concisely presenting the experimental results first followed by DFT discussion. Besides, the authors should provide a clear link between experiments and DFT calculations:

- (i) Hot electrons are observed in experiments. How are these hot electrons included in the DFT calculations? This needs to be discussed clearly.
- (ii) Pt(111) has been used as a model for Pt/thinfilms calculations. What is the thickness of Pt layers on TiO₂? Does the Pt(111) surface used here reasonably represent Pt/TiO₂ film in experiment? For example it is known that Pt monolayer, bilayer/substrate configurations show very different activity and selectivity compared to the Pt(111) surface.
- (iii) The authors picked certain steps to explain the reactivity difference in Figure 4. It is essential to compare the full reaction channel: from reactant (CH₃OH) to products to make final conclusion about the selectivity difference between two catalysts.

Reply to Reviewer 3's report

Referee comment #0: *The authors have added additional experimental and theoretical results to signify the role of metal-oxide interface. Nevertheless, as the previous reviewer mentioned there is significant room to improve the organization of the manuscript: for example concisely presenting the experimental results first followed by DFT discussion.*

Reply #0: We thank the reviewer for their interest and valuable comments. As the reviewer pointed out, we needed to show the experimental result parts more concisely in our manuscript. We reorganized the whole manuscript to improve the readability. Reflecting on what the reviewer pointed out, in order to concisely present our experimental results first, we moved most of the contents that do not need to be emphasized in our previous main manuscript to Supplementary Notes (see **Supplementary Notes 1-7 in Supplementary Information**), further emphasizing the content that supports the important claims in this study (i.e., enhancement of selectivity and hot electron excitation on metal-oxide interfaces). We feel that the manuscript has not been clearly presented and have thereby revised the manuscript extensively to improve the clarity of our claim. Detailed responses to the reviewer's comments are given below.

Referee comment #1. *Hot electrons are observed in experiments. How are these hot electrons included in the DFT calculations? This needs to be discussed clearly.*

Reply #1: We thank the reviewer for this meticulous comment regarding the theoretical calculations of hot electron transfer. However, since hot electrons generated by non-adiabatic electronic excitation by exothermic catalytic reaction are *excited* states, it is challenging to directly calculate the transfer of excited hot electrons through DFT calculation and compare them in the two catalysts (i.e., Pt film/TiO₂ and Pt nanowires/TiO₂). However, in our earlier DFT studies, we were able to prove the relationship between the generation of methyl formate and hot electron generation obtained experimentally in the Pt model system through theoretical DFT calculations. In other words, since hot electrons were generated by exothermic surface reaction, when the exothermic energy in the pathway where CO₂ and methyl formate were calculated through DFT calculation, the exothermicity of methyl formate production was higher than CO₂ formation, resulting in higher hot electron generation (Lee, S. W. *et al. ACS Catal.* **9**, 8424-8432 (2019)). In addition to the previous studies, in our current studies, it was experimentally found that the selectivity of methyl formate formation was further enhanced when the Pt/TiO₂ interface was formed, and this fact was proved through the difference of the activation barrier in the pathway of each product generation in Pt(111) and Pt nanorod/TiO₂(110) through our DFT calculation (**Fig. 4**). Hence,

it was found that the *increase in selectivity at the formed interface further promoted reaction-induced hot electron generation*. In this study, hot electrons were not treated explicitly in DFT calculation, but by means of energetic comparison between the selectivity of methyl formate and CO₂ on Pt-TiO₂ and that of Pt(111) via DFT calculations, the enhanced hot electron generation on Pt-TiO₂ interface was rationalized. In response to comment, we added these descriptions to the revised manuscript.

Referee comment #2. *Pt(111) has been used as a model for Pt/thin films calculations. What is the thickness of Pt layers on TiO₂? Does the Pt(111) surface used here reasonably represent Pt/TiO₂ film in experiment? For example it is known that Pt monolayer, bilayer/substrate configurations show very different activity and selectivity compared to the Pt(111) surface.*

Reply #2: We are grateful that the reviewer commented on the thickness of the Pt film on TiO₂. The thickness of the Pt film was 5 nm in the Pt film/TiO₂, which is close to bulk Pt, so we can conclude that the effect of the substrate is negligible. In addition, as confirmed in the SEM image (**Supplementary Fig. 3d**), this 5 nm-thick Pt film was uniformly deposited on TiO₂, so that TiO₂ substrate was not exposed to reaction gas. That is, in the Pt film/TiO₂ catalyst, only the Pt *surface* participated in the catalytic reaction.

In addition, to further confirm the substrate effect discussed by the reviewer, we compared the selectivity after depositing the Pt film with the same thickness of 5 nm on a SiO₂ substrate instead of TiO₂. As shown in **Fig. R1**, it was confirmed that the selectivity of Pt film/TiO₂ and Pt film/SiO₂ were similar in all methanol conversions, and thus it was confirmed once again that the 5 nm-thick Pt film had no substrate effect. Hence, we can confirm that the *three-atomic-layer (3×3) of Pt(111) surface unit cell* with a fixed bottom layer in their bulk position we used as a model in the DFT calculation reasonably represents the *5 nm-thick Pt film on TiO₂ substrate*. We are grateful that the reviewer pointed out the oversight, and we added these explanations to the revised manuscript (**Supplementary Note 3**).

Fig. R1 | Plot of selectivity according to methanol conversion on the Pt film/TiO₂ and Pt film/SiO₂ measured at 343 K.

Referee comment #3. *The authors picked certain steps to explain the reactivity difference in Figure 4. It is essential to compare the full reaction channel: from reactant (CH₃OH) to products to make final conclusion about the selectivity difference between two catalysts.*

Reply #3: Thank you for this valuable insight. As previously reported in other literature (Wittstock, A. *et al.*, *Science* **327**, 319-322 (2010)), in methanol oxidation, methanol is first oxidized to generate methoxy (*CH₃O), and then the formaldehyde intermediate (*HC(=O)H) is produced by additional deprotonation. As shown in **Fig. R2** or **Supplementary Fig. 24**, when the formaldehyde intermediate reacts with methoxy C-C coupling occurs, methyl formate is produced. On the other hand, when this intermediate reacts with oxygen and undergoes oxidation, CO₂ is produced. In other words, the reaction of the formaldehyde intermediate determines which is produced between methyl formate and CO₂ (Personick, M. L. *et al.*, *ACS Catal.* **7**, 965-985 (2017)). Hence, to compare the *selectivity* in the methanol oxidation reaction, it was possible to compare the reaction channel from formaldehyde in two catalysts.

Furthermore, as pointed out by the reviewer, we additionally calculated all of the activation barriers in all possible reaction pathways from reactant to two products when the methanol oxidation reaction occurs in Pt nanorod/TiO₂(110) and Pt(111) (see **Supplementary Tables 4 and 5** below). Note that, in **Fig. 4**, we picked the *most energetically favorable steps* among the possible steps from reactant to both products. Since two reactions split off from formaldehyde (CH₂O), the steps from methanol to formaldehyde have been omitted in **Fig. 4**. We consider all the activation barriers, but the main theoretical trends (*i.e.*, barrier differences in reaction determining steps on two models) still remains the same as before. Reflecting the parts pointed out by the reviewer, we additionally calculated activation energy and

exothermic energy in all pathways from reactant to final product, and added **Supplementary Tables 4 and 5**.

Fig. R2 | Two reaction pathways for the production of methyl formate and CO₂ from methanol.

Supplementary Fig. 24 | Scheme of the reaction paths for methanol oxidation to **a** CH₂O intermediate, **b** methyl formate and **c** CO₂ formation.

	reaction pathway	reaction energy (eV)	barrier (eV)
CH ₃ OH to form CH ₃ O			
1	*CH ₃ OH → *CH ₃ O+*H	0.06	1.10
2	*CH ₃ OH+*O → *CH ₃ O+*OH	-0.35	0.24
CH ₃ O to form CH ₂ O			
3	*CH ₃ O → *CH ₂ O + *H	0.40	0.59
4	*CH ₃ O+*O → *CH ₂ O + *OH	-0.38	0.56
5	*CH ₃ O+*OH → *CH ₂ O + *H ₂ O	-0.72	0.08
6	*CH ₃ O+*CH ₃ O → *CH ₂ O + *CH ₃ OH	-0.25	0.73
Formation of alkoxy hemiacetal			
7	*CH ₃ O + *CH ₂ O → *CH ₃ OCOH ₂	-0.07	0.00
Alkoxy hemiacetal to form methylformate			
8	*CH ₃ OCOH ₂ → CH ₃ OCOH _(g) + *H	0.08	0.84
9	*CH ₃ OCOH ₂ + *O → CH ₃ OCOH _(g) + *OH	-0.44	0.55
10	*CH ₃ OCOH ₂ + *OH → CH ₃ OCOH _(g) + *H ₂ O	-1.45	0.16
CH ₂ O to form HCO			
11	*CH ₂ O → *HCO + *H	0.02	0.63
12	*CH ₂ O + *O → *HCO + *OH	-0.78	1.07
13	*CH ₂ O+ *OH → *HCO + *H ₂ O	-1.48	0.56

HCO to form CO			
14	$*\text{HCO} \rightarrow *\text{CO} + *\text{H}$	-0.05	0.48
15	$*\text{HCO} + *\text{O} \rightarrow *\text{CO} + *\text{OH}$	-1.25	0.56
16	$*\text{HCO} + *\text{OH} \rightarrow *\text{CO} + *\text{H}_2\text{O}$	-1.73	0.59
Formation of CO ₂			
17	$*\text{CO} + *\text{O} \rightarrow \text{CO}_2$	-1.54	1.24

Supplementary Table 4 | Calculated reaction energies and barriers for possible reaction pathways for methanol oxidation on Pt nanorod on TiO₂(110).

	reaction pathway	reaction energy (eV)	barrier (eV)
CH ₃ OH to form CH ₃ O			
1	$*\text{CH}_3\text{OH} \rightarrow *\text{CH}_3\text{O} + *\text{H}$	0.20	0.65
2	$*\text{CH}_3\text{OH} + *\text{O} \rightarrow *\text{CH}_3\text{O} + *\text{OH}$	0.26	0.28
CH ₃ O to form CH ₂ O			
3	$*\text{CH}_3\text{O} \rightarrow *\text{CH}_2\text{O} + *\text{H}$	-0.60	0.20
4	$*\text{CH}_3\text{O} + *\text{O} \rightarrow *\text{CH}_2\text{O} + *\text{OH}$	-0.38	0.81
5	$*\text{CH}_3\text{O} + *\text{OH} \rightarrow *\text{CH}_2\text{O} + *\text{H}_2\text{O}$	-1.25	0.42
6	$*\text{CH}_3\text{O} + *\text{CH}_3\text{O} \rightarrow *\text{CH}_2\text{O} + *\text{CH}_3\text{OH}$	-1.16	0.51
Formation of alkoxy hemiacetal			
7	$*\text{CH}_3\text{O} + *\text{CH}_2\text{O} \rightarrow *\text{CH}_3\text{OCOH}_2$	-0.06	0.64
Alkoxy hemiacetal to form methylformate			
8	$*\text{CH}_3\text{OCOH}_2 \rightarrow \text{CH}_3\text{OCOH}_{(\text{g})} + *\text{H}$	-1.75	1.89
9	$*\text{CH}_3\text{OCOH}_2 + *\text{O} \rightarrow \text{CH}_3\text{OCOH}_{(\text{g})} + *\text{OH}$	-1.48	0.00
10	$*\text{CH}_3\text{OCOH}_2 + *\text{OH} \rightarrow \text{CH}_3\text{OCOH}_{(\text{g})} + *\text{H}_2\text{O}$	-2.56	0.00
CH ₂ O to form HCO			
11	$*\text{CH}_2\text{O} \rightarrow *\text{HCO} + *\text{H}$	-0.70	0.17
12	$*\text{CH}_2\text{O} + *\text{O} \rightarrow *\text{HCO} + *\text{OH}$	-0.99	0.25
13	$*\text{CH}_2\text{O} + *\text{OH} \rightarrow *\text{HCO} + *\text{H}_2\text{O}$	-1.79	0.00

HCO to form CO			
14	$*\text{HCO} \rightarrow *\text{CO} + *\text{H}$	-1.05	0.25
15	$*\text{HCO} + *\text{O} \rightarrow *\text{CO} + *\text{OH}$	-1.05	0.03
16	$*\text{HCO} + *\text{OH} \rightarrow *\text{CO} + *\text{H}_2\text{O}$	-1.68	0.32
Formation of CO ₂			
17	$*\text{CO} + *\text{O} \rightarrow \text{CO}_2$	-0.61	0.86

Supplementary Table 5 | Calculated reaction energies and barriers for possible reaction pathways for methanol oxidation on Pt(111).

REVIEWERS' COMMENTS

Reviewer #3 (Remarks to the Author):

The authors performed additional DFT calculations which help clarify the preferred mechanisms proposed in the manuscript. I recommend the publication of this work in Nature Communications.

In page 19: "...the binding energy of CO₂ on the TiO₂ surface was 0.21 eV" . The binding energy should be -0.21 eV.

Reply to Reviewer 3's report

General comment: *The authors performed additional DFT calculations which help clarify the preferred mechanisms proposed in the manuscript. I recommend the publication of this work in Nature Communications.*

In page 19: "...the binding energy of CO₂ on the TiO₂ surface was 0.21 eV" . The binding energy should be -0.21 eV.

Reply: We appreciate the referee's recommendation to publish our revised manuscript in *Nature Communications*. Also, thank you for the reviewer's finding and pointing out the typo in our revised manuscript. As suggested by the reviewer, we modified the direction of binding energy to be correct. We truly appreciate reviewer's time and effort on our manuscript.